# Prolyl hydroxylase substrate adenylosuccinate lyase is an oncogenic driver in triple negative breast cancer

Giada Zurlo[1], Xijuan Liu[1], Mamoru Takada[1], Cheng Fan[1], Jeremy M. Simon [1,2,3], Travis S. Ptacek[1,3], Javier Rodriguez[4], Alex von Kriegsheim[4], Juan Liu[5], Jason W. Locasale [5], Adam Robinson[1], Jing Zhang [1,6], Jessica M. Holler[7], Baek Kim [7], Marie Zikánová[8], Jörgen Bierau[9], Ling Xie[10], Xian Chen[10], Mingjie Li[1], Charles M. Perou [1] & Qing Zhang [1,6,11,12]*

Protein hydroxylation affects protein stability, activity, and interactome, therefore contributing to various diseases including cancers. However, the transiency of the hydroxylation reaction hinders the identification of hydroxylase substrates. By developing an enzyme-substrate trapping strategy coupled with TAP-TAG or orthogonal GST- purification followed by mass spectrometry, we identify adenylosuccinate lyase (ADSL) as an EglN2 hydroxylase substrate in triple negative breast cancer (TNBC). ADSL expression is higher in TNBC than other breast cancer subtypes or normal breast tissues. ADSL knockout impairs TNBC cell proliferation and invasiveness in vitro and in vivo. An integrated transcriptomics and metabolomics analysis reveals that ADSL activates the oncogenic cMYC pathway by regulating cMYC protein level via a mechanism requiring ADSL proline 24 hydroxylation. Hydroxylation-proficient ADSL, by affecting adenosine levels, represses the expression of the long noncoding RNA *MIR22HG*, thus upregulating cMYC protein level. Our findings highlight the role of ADSL hydroxylation in controlling cMYC and TNBC tumorigenesis.

[1] Lineberger Comprehensive Cancer Center, University of North Carolina School of Medicine, Chapel Hill, NC 27599, USA. [2] Department of Genetics, Neuroscience Center, University of North Carolina, Chapel Hill, NC 27599, USA. [3] UNC Neuroscience Center, Carolina Institute for Developmental Disabilities, University of North Carolina, Chapel Hill, NC 27599, USA. [4] Cancer Research UK Edinburgh Centre, IGMM, University of Edinburgh, Edinburgh EH4 2XR, UK. [5] Department of Pharmacology and Cancer Biology, Duke University School of Medicine, Durham, NC 27710, USA. [6] Department of Pathology and Laboratory Medicine, University of North Carolina, Chapel Hill, NC 27599, USA. [7] Department of Pediatrics, School of Medicine, Emory University, Atlanta, GA 30322, USA. [8] Research Unit for Rare Diseases, Department of Pediatrics and Adolescent Medicine, First Faculty of Medicine, Charles University and General University Hospital in Prague, Prague, Czechia. [9] Department of Clinical Genetics, Maastricht University Medical Centre, Maastricht, The Netherlands. [10] Department of Biochemistry and Biophysics, University of North Carolina, Chapel Hill, NC 27599, USA. [11] Department of Pharmacology, University of North Carolina, Chapel Hill, NC 27599, USA. [12] Department of Pathology, UT Southwestern Medical Center, Dallas, TX 75390, USA. *email: Qing.Zhang@utsouthwestern.edu

Triple negative breast cancer (TNBC) is the breast cancer subtype characterized by the absence of expression of the estrogen and progesterone receptors, as well as amplification of HER2. Therapies commonly used in other breast cancer subtypes are therefore not suitable for TNBC, and treatment options are largely limited to conventional genotoxic chemotherapy[1]. Most of TNBC patients present residual disease after chemotherapy, with high rates of metastatic recurrence and very poor long-term prognosis[2]. Consequently, the identification of molecular targets and the development of new therapeutic avenues remain critically important.

We previously showed the hydroxylase EglN2 positively contributing to TNBC in vitro and in vivo[3]. However, the underlying mechanism was unknown. Here, we show that EglN2 hydroxylase activity is important for mediating the phenotype of EglN2 in TNBC, and EglN2 knockdown does not promote any increase in the protein level of HIF-1α, its known substrate. These observations led to the hypothesis that EglN2 oncogenic behavior in TNBC is mediated by its hydroxylation of another substrate. Detecting hydroxylation substrates is challenging due to the transiency of the reaction: upon hydroxylation, the substrate is quickly released by the hydroxylase. To overcome this issue, we use an optimized substrate-trapping strategy[4], which fixes the enzyme-substrate complex in an inactive state, thus increasing the amount of the substrate bound to the hydroxylase. To avoid the possibility that the predominant substrate HIF-1α, along with its binding partners, may overshadow the other EglN2 substrates, we use HIF-1α-depleted cells. Our screening combines the optimized substrate-trapping strategy with a TAP-TAG purification followed by mass spectrometry. In parallel, we develop an orthogonal approach by combining this enzyme-substrate-trapping strategy to GST-EglN2 pull-down followed by mass spectrometry. Adenylosuccinate lyase (ADSL) is the only substrate identified by both approaches, giving us confidence on ADSL being a bona fide EglN2 substrate. The ADSL relationship with EglN2, as well as its role in TNBC, is then investigated.

## Results

**Substrate-trapping strategy shows ADSL as an EglN2 substrate.** Prolyl hydroxylation is a transient reaction characterized by the binding of the EglN prolyl hydroxylase to its cofactors, $Fe^{2+}$ and α-ketoglutarate, its substrate and oxygen. Once the hydroxylation reaction takes place, the enzyme and substrate immediately dissociate, which makes the detection of the prolyl hydroxylase substrates by regular pull-down and mass spectrometry very challenging. Previous research showed that by treating cells with dimethyloxaloylglycine (DMOG), a competitive antagonist of α-ketoglutarate and hydroxylase inhibitor, the potential hydroxylase substrates may be trapped within the reaction complex and, therefore, more easily detectable[4]. However, the modest number of binding proteins pulled down from mass spectrometry suggests that there may be other ways to enrich for the prolyl hydroxylase substrates. The addition of oxygen triggers prolyl hydroxylation of the substrates, which dissociate from the prolyl hydroxylase. We reasoned that limited oxygen concentration might help trap the enzyme-substrate interaction. Interestingly, we did not find that hypoxia treatment alone traps EglN2 interaction with its known substrate HIF-1α in two $ER^{+}$ breast cancer cell lines, T47D and MCF7 (Fig. 1a and Supplementary Fig. 1a). However, we found that the combinatorial treatment Hypoxia + DMOG produced a robust binding (Fig. 1a and Supplementary Fig. 1a). This was not solely due to the upregulation of HIF-1α under this condition. In fact, under the same condition, catalytically dead EglN2 (H358A) could not bind with HIF-1α (Fig. 1a). Moreover,

despite accumulating comparable amounts of HIF-1α, hypoxia or hypoxia + the proteasome inhibitor MG132 did not induce any distinctive EglN2-HIF-1α binding (Fig. 1a and Supplementary Fig. 1a). To further validate our EglN2-substrate interaction enrichment strategy, we performed a similar experiment in the Triple Negative Breast Cancer (TNBC) cell line MDA-MB-231. Again, hypoxia + DMOG treatment gave rise to a strong binding between the WT (but not the H358A mutant) EglN2 prolyl hydroxylase and its substrate HIF-1α (Fig. 1b).

In a previous study, we demonstrated that EglN2 prolyl hydroxylase contributes to TNBC tumor progression[3]. Consistently, we show here that EglN2 knockdown by shRNA (Supplementary Fig. 1b) decreases TNBC tumor cell invasion, the phenotype being rescued by WT, but not the catalytically dead mutant, EglN2 (Supplementary Fig. 1c). However, the underlying mechanism remains unclear. EglN2 depletion does not lead to HIF-1α stabilization (Supplementary Fig. 1d), suggesting that there may be other EglN2 substrates contributing to its effect in TNBC. To further enrich for the non-HIF substrates that may be overshadowed by the presence of HIF-1α, we depleted HIF-1α in MDA-MB-231 cells (Fig. 1c). Then, we performed our optimized trapping strategy treating these HIF-1α-depleted cells, previously transduced with FLAG-HA-WT EglN2, with either normoxia or hypoxia + DMOG, followed by TAP-TAG purification with FLAG and HA affinity gels (Fig. 1c). In parallel, we performed the pull-down experiments using GST-control or -EglN2 beads (Fig. 1c). The mass spectrometry analysis that followed both approaches highlighted a handful of proteins showing a stronger binding with EglN2 in hypoxia + DMOG compared with normoxia (Supplementary Tables 1 and 2). Particularly, adenylosuccinate lyase (ADSL) was retrieved from both mass spectrometry experiments as the only common EglN2 interaction protein (Fig. 1c, Supplementary Tables 1 and 2), encouraging us to pursue the validation of it as a potential EglN2 substrate.

To confirm ADSL interaction with EglN2, we performed a GST-Ctrl or -EglN2 pull-down in MDA-MB-231 cells: ADSL could be precipitated by GST-EglN2 in the substrate-trapping condition (hypoxia + DMOG) (Fig. 1d). We then immunoprecipitated EglN2-WT or -H358A in MDA-MB-231 cells. The binding of wild-type EglN2 with ADSL was much stronger than that of the catalytically inactive H358A mutant, suggesting that not only is ADSL an interactor, but also a substrate, of EglN2 (Fig. 1e). We verified that their interaction is physiological by reciprocal immunoprecipitations of endogenous EglN2 or ADSL: endogenous ADSL or EglN2 was retrieved, respectively (Fig. 1f). To examine whether the binding between EglN2 and ADSL is direct, we used two complementary in vitro approaches. We pulled down in vitro translated ADSL with GST-EglN2, and found an interaction between these two proteins (Fig. 1g). Conversely, we performed GST-ADSL pull-down with FLAG-EglN2 recombinant protein purified from insect cells and observed a direct interaction (Fig. 1h). Collectively, our data suggest that EglN2 and ADSL interact without requiring additional partners. To determine whether the other members of the EglN family, EglN1 and 3, may also bind with ADSL, we repeated the GST pull-down experiments: EglN2 is the primary prolyl hydroxylase binding ADSL (Fig. 1i). Importantly, we assessed whether ADSL is one of the key mediators of EglN2 depletion phenotype in TNBC by over-expressing ADSL in EglN2-depleted MDA-MB-231 cells (Supplementary Fig. 1e). ADSL overexpression, even if moderate, was able to partially restore MDA-MB-231 2-D colony formation (Supplementary Fig. 1f), suggesting that the oncogenic role of EglN2 in TNBC may be at least partially mediated by ADSL.

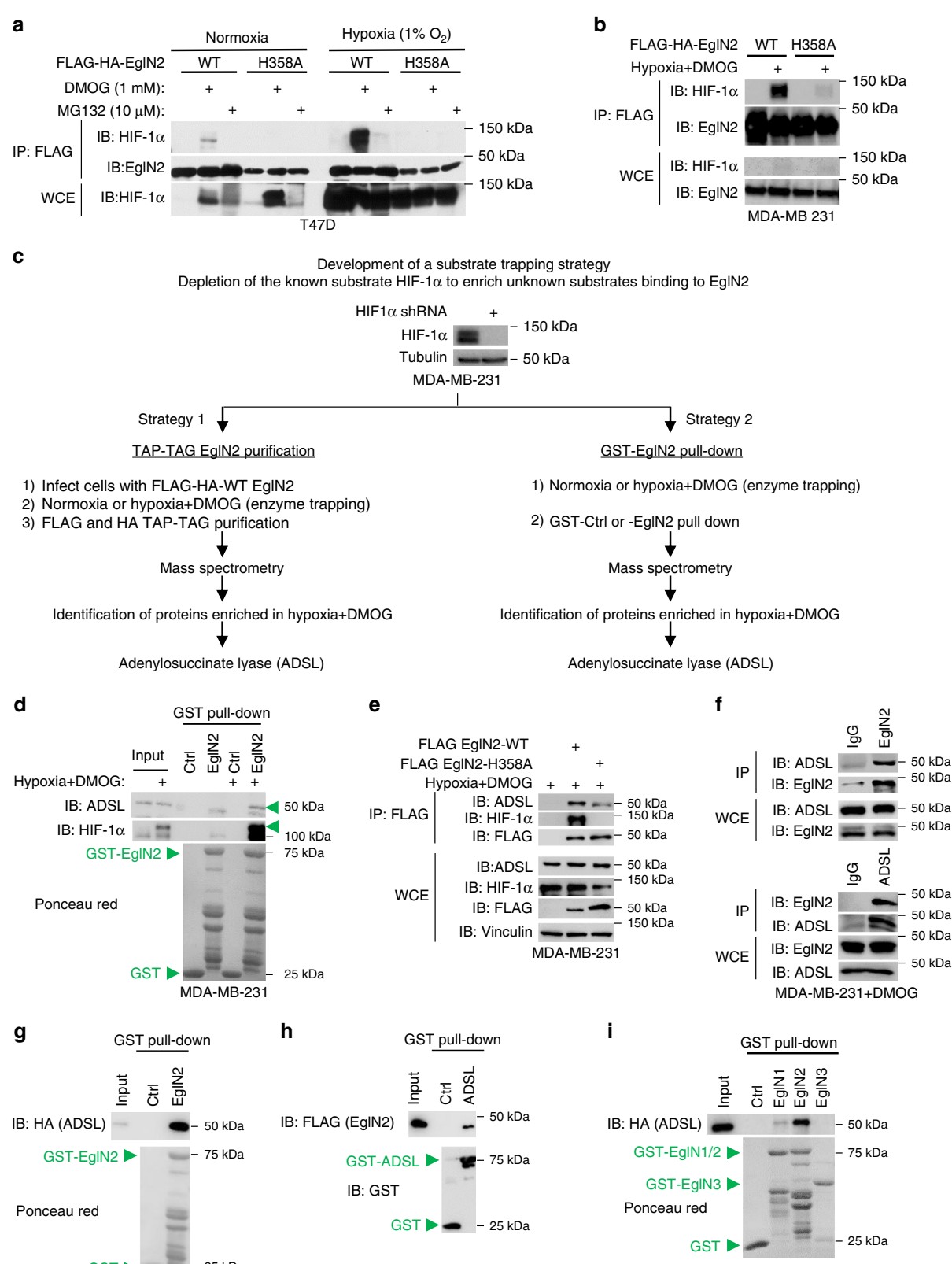

**ADSL is essential for TNBC cell growth and invasiveness.** ADSL is a key enzyme involved in purine metabolism that catalyzes two non-sequential reactions in the purine biosynthetic pathway[5]. The first reaction converts succinylaminoimidazole carboxamide ribotide (SAICAR) to aminoimidazole carboxamide ribotide (AICAR) and the second adenylosuccinate (SAMP) to adenosine monophosphate (AMP). Both reactions also produce fumarate, which is a metabolite involved in the tricarboxylic acid cycle and an epigenetic modifier promoting epithelial-to-mesenchymal transition[6]. However, the role of ADSL in cancer, especially TNBC, remains elusive. To study ADSL expression in breast cancer patients, we examined ADSL mRNA level in different subtypes of breast cancer across different datasets: The Cancer Genome Atlas (TCGA)[7], UNC337[8], and METABRIC[9].

**Fig. 1** Substrate-trapping strategy identifies ADSL as an EglN2 substrate in TNBC. **a** Immunoblots (IB) of whole-cell extracts (WCE) and immunoprecipitations (IP) of lysates from T47D cells infected with lentivirus encoding either wild-type (WT) or catalytically dead mutant (H358A) FLAG- and HA-tagged EglN2, and treated as indicated overnight. **b** IB of WCE and IP of lysates from MDA-MB-231 cells infected with lentivirus encoding either EglN2-WT or -H358A, and treated as indicated overnight. **c** Schematic representation of the hydroxylase (here EglN2) substrate screen. **d** IB of input (1% of the protein amount used for the pull-down) and GST pull-down of lysates from MDA-MB-231 cells treated as indicated overnight. **e** IB of WCE and IP of lysates from MDA-MB-231 cells transfected with indicated plasmids and treated as indicated overnight. **f** Endogenous EglN2 and ADSL IP in MDA-MB-231 treated with DMOG. **g, h, i** IB of input and GST pull-down of in vitro-translated ADSL (**g**, **i**) and recombinant EglN2 (**h**). In **b**, **c**, **d**, and **e**, hypoxia was 1% $O_2$ and DMOG concentration was 1 mM. Source data are provided as a Source Data file

ADSL displayed the highest expression in TNBC compared with the other subtypes of breast cancer, i.e., Her2[+], luminal (A and B), and normal-like (Fig. 2a). Importantly, ADSL expression in TNBC was higher than in the normal breast tissues (Fig. 2b). We also assayed ADSL protein levels in TNBC compared with normal breast tissues derived from nine patients. Despite the variation between different normal and tumor samples, likely reflecting patient tissue heterogeneity, ADSL was upregulated in tumors compared with their adjacent normal tissue (Fig. 2b). By performing Kaplan-Meier curves, COX Proportional Hazard Model and Concordance Index in TCGA, UNC337, and METABRIC, no correlation was found between ADSL mRNA level and TNBC patient survival (Supplementary Fig. 2a). In this regard, we would like to underline that the prognosis data are neither necessary nor sufficient to predict whether the target gene is a good therapeutic target in cancer[10].

In order to examine the potential role of ADSL in TNBC, we implemented CRISPR-Cas9 to deplete ADSL expression in the TNBC cell lines MDA-MB-231, MDA-MB-436, and MDA-MB-468. ADSL knockout by using two independent sgRNAs impaired breast cancer cell proliferation (Fig. 2c). We depleted ADSL in the normal breast epithelial cell lines MCF-10A and HMLE cells: the growth defect observed was not as dramatic as that in TNBC cell lines (Fig. 2c). ADSL depletion also affected TNBC anchorage-independent growth in cells, as shown by decreased soft-agar colony formation (Fig. 2d). Consistently, ADSL knockout cells displayed an impaired invasiveness (Fig. 2e). In accordance with ADSL sgRNA data, ADSL downregulation via two independent shRNAs (Supplementary Fig. 2b) led to decreased soft-agar growth (Supplementary Fig. 2c) and invasiveness (Supplementary Fig. 2d). To confirm that the effect of ADSL sgRNAs on TNBC phenotype was on-target, we generated an MDA-MB-231 cell line expressing the doxycycline (dox)-inducible 3xFLAG-tagged ADSL (with CRISPR-Cas9 NGG recognition sequence silently mutated) to a level comparable to the endogenous ADSL. We then transduced this cell line with the lentivirus expressing the ADSL sgRNA (Fig. 2f). In the absence of doxycycline, ADSL knockout led to decreased soft-agar growth and invasion (Fig. 2g, h). In the presence of doxycycline, the exogenous ADSL rescued the defective phenotype elicited by ADSL depletion (Fig. 2g, h). We used two other TNBC cell lines, MDA-MB-436 and MDA-MB-468, and found that ADSL depletion in these cells also led to decreased soft-agar growth (Supplementary Fig. 2e) and cell invasion (Supplementary Fig. 2f). Similar to what we observed for MDA-MB-231, when the exogenous ADSL was re-introduced into these cells (Supplementary Figs. 2g and i), the 3-D cell growth defect and decreased cell invasion were rescued (Supplementary Figs. 2h and j). Cumulatively, our data suggest that ADSL contributes to TNBC cell proliferation, anchorage-independent growth, and cell invasion.

To examine the role of ADSL in vivo, we orthotopically injected MDA-MB-231 cells transduced with either control- or ADSL sgRNA-expressing lentivirus into the mammary fat pad of NOD SCID Gamma (NSG) mice. These cells were labeled with luciferase to monitor tumor growth in vivo by using weekly bioluminescence as described previously[11]. It is worth noting that the bioluminescence intensity at week 1 was comparable between control and ADSL sgRNA, indicating comparable tumor cell implantation (Fig. 3a). Overtime, differently from control cells, ADSL-depleted tumor cells failed to grow (Fig. 3a, b), which was reflected by the decreased tumor size and weight at necropsy (Fig. 3c). To further determine whether ADSL may be important for TNBC metastasis, we injected these cells into NSG mice through the tail vein. Consistent with the phenotype observed with primary tumor growth, ADSL depletion diminished tumor cell colonization in the lung (Fig. 3d–f), even though the starting amount of injected cells was comparable (Fig. 3e). We observed a similar phenotype using another independent sgRNA targeting ADSL (Supplementary Figs. 3a and b). Moreover, the mammary fat pad orthotopic injection of another triple negative cell line (MDA-MB-468) also displayed a growth defect in the tumors generated by ADSL-depleted cells (Supplementary Figs. 3c and d). Therefore, ADSL contributes to TNBC breast tumorigenesis and metastasis in vivo.

**ADSL is hydroxylated by EglN2 on Proline 24**. To identify the specific ADSL proline residues hydroxylated by EglN2, we performed in vitro hydroxylation of GST-ADSL with recombinant EglN2: EglN2 promoted GST-ADSL prolyl hydroxylation on Proline 24 (Fig. 4a, b). Conversely, we transfected cells with HA-tagged-ADSL followed by MG132 or MG132 + DMOG treatment. DMOG treatment decreased the ADSL hydroxylation on Proline 24 (Supplementary Fig 4a). We performed pan-hydroxyproline IP after EglN2 overexpression in 293T cells. The ADSL blot that followed confirmed that endogenous ADSL is hydroxylated in cells by EglN2, and this hydroxylation is impaired by the hydroxylase inhibitor DMOG (Fig. 4c). To further confirm proline 24 crucial role for ADSL hydroxylation, we mutated it to alanine (P24A). Then, we overexpressed EglN2 and either WT or P24A mutant ADSL in 293T cells, followed by pan-hydroxyproline IP. The hydroxylation signal detected for WT ADSL was lost for P24A mutant, phenocopying the effect of DMOG treatment (Fig. 4d). We also performed an in vitro hydroxylation reaction of eluted GST-ADSL (WT and P24A) in the presence or absence of recombinant EglN2, followed by IP with pan-hydroxyproline antibody. Consistently, we observed an increase in the hydroxylation signal of WT, but not P24A, ADSL in the presence of recombinant EglN2 (Supplementary Fig 4b). Therefore, our results suggest that ADSL Proline 24 is the major site hydroxylated by EglN2.

We showed that EglN2 can hydroxylate ADSL on proline 24 in vitro and in cells. However, questions remained about the physiological relevance of ADSL hydroxylation. ADSL protein stability was not affected by this posttranslational modification, since ADSL protein levels did not change upon Hypoxia + DMOG treatment (Supplementary Fig. 4c) or EglN2 depletion (Supplementary Fig. 4d). To address the question of the potential relevance of ADSL hydroxylation, we re-introduced either WT, P24A or the catalytically impaired ADSL mutant R426H[5] in

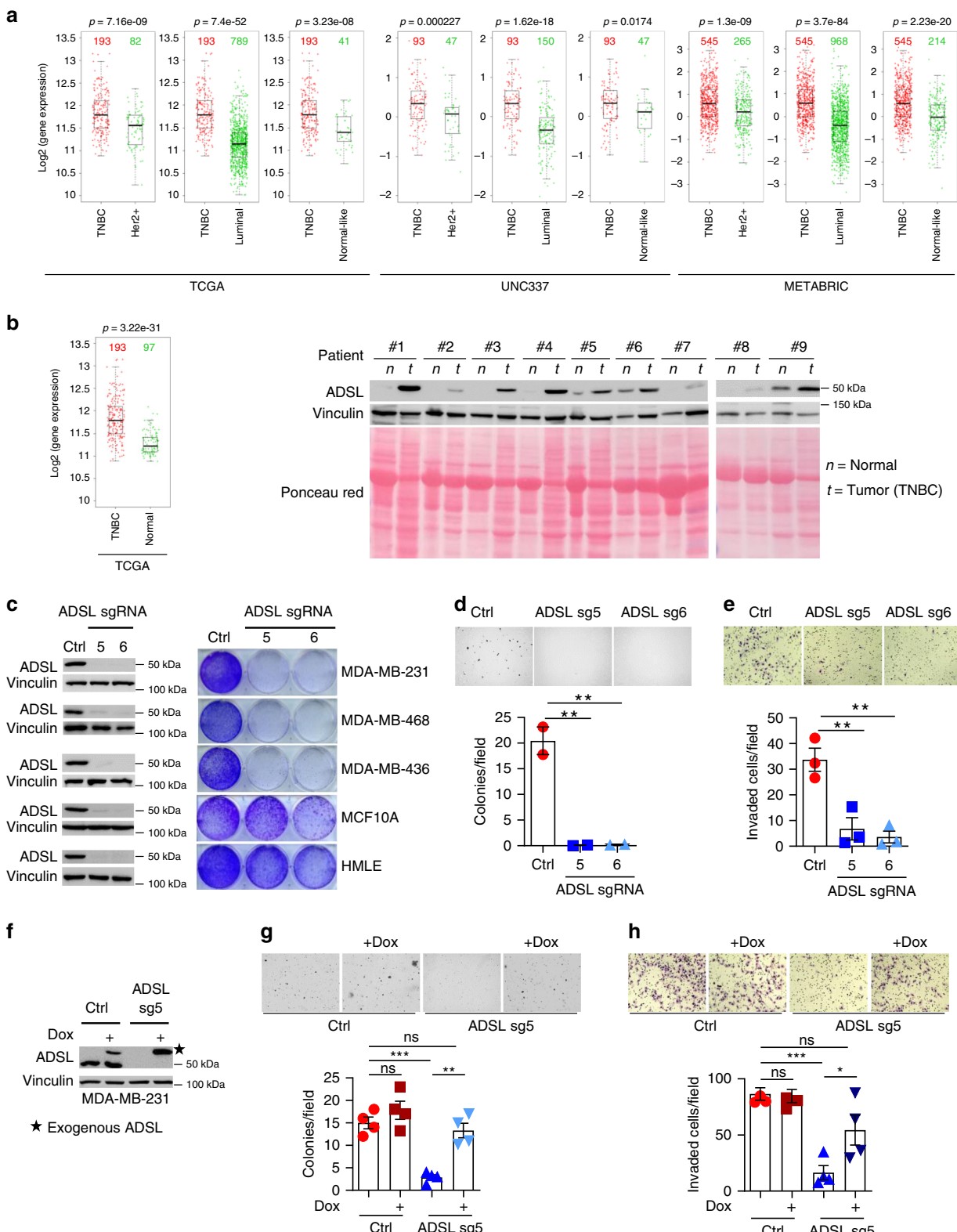

ADSL knockout cells. WT ADSL could rescue the ADSL-depleted cell proliferation. Conversely, despite an expression comparable to the WT or endogenous ADSL (Fig. 4e), both P24A and R426H failed to fully rescue the phenotype (Fig. 4f), suggesting that ADSL proline 24 residue may affect ADSL activity, which would be important for the ADSL depletion phenotype in TNBC cells. To confirm this hypothesis, we measured ADSL activity using both substrates, SAMP and SAICAR. As expected, R426H showed

a reduced activity compared with WT ADSL (Fig. 4g). Importantly, P24A mutant activity was significantly reduced (Fig. 4g). We assessed WT and P24A ADSL activity by an orthogonal approach, by tracking the labeling of AMP, product of ADSL-catalyzed reaction, from [U-$^{13}$C$_6$]glucose. The isotope incorporation into AMP was restored to a level comparable to the control in ADSL-depleted cells expressing WT, but not P24A, ADSL (Supplementary Fig 4e). Collectively, these results suggest that

**Fig. 2** ADSL plays an essential role in TNBC. **a** ADSL mRNA expression across different subtypes of breast cancer in three different datasets (TCGA stands for The Cancer Genome Atlas). The center line of the dox plots represents the median, the bounds the upper and lower quartiles, and the whiskers the 1.5× interquartile range. **b** ADSL mRNA expression in TNBC and normal breast tissue in TCGA, and IB of lysates from paired TNBC patient-derived non-tumor (n) and tumor (t) breast tissues. **c** Representative 2-D proliferation of indicated cells upon ADSL depletion shown by IB. **d, e** Representative images of (**d**) anchorage-independent growth and (**e**) invasion of MDA-MB-231 cells infected with either ADSL sgRNAs #5 and #6 (sg5 and sg6) or sgRNA control (Ctrl). Graphs represent the mean ± SEM from two independent experiments, each performed in duplicate (**d**), and from three independent experiments (**e**). **P < 0.01 were calculated using one-way ANOVA followed by Dunnett's multiple comparison test. **f** IB of lysates from MDA-MB-231 cells overexpressing doxycycline (dox)-inducible ADSL, infected with indicated virus and treated as indicated. **g, h** Representative images of (**g**) anchorage-independent growth and (**h**) invasion of MDA-MB-231 cells overexpressing dox-inducible ADSL, infected and treated as indicated. Graphs represent the mean ± SEM from four independent samples. *P < 0.05, **P < 0.01, ***P < 0.001 were calculated using one-way ANOVA followed by Tukey's multiple comparison test. Source data are provided as a Source Data file

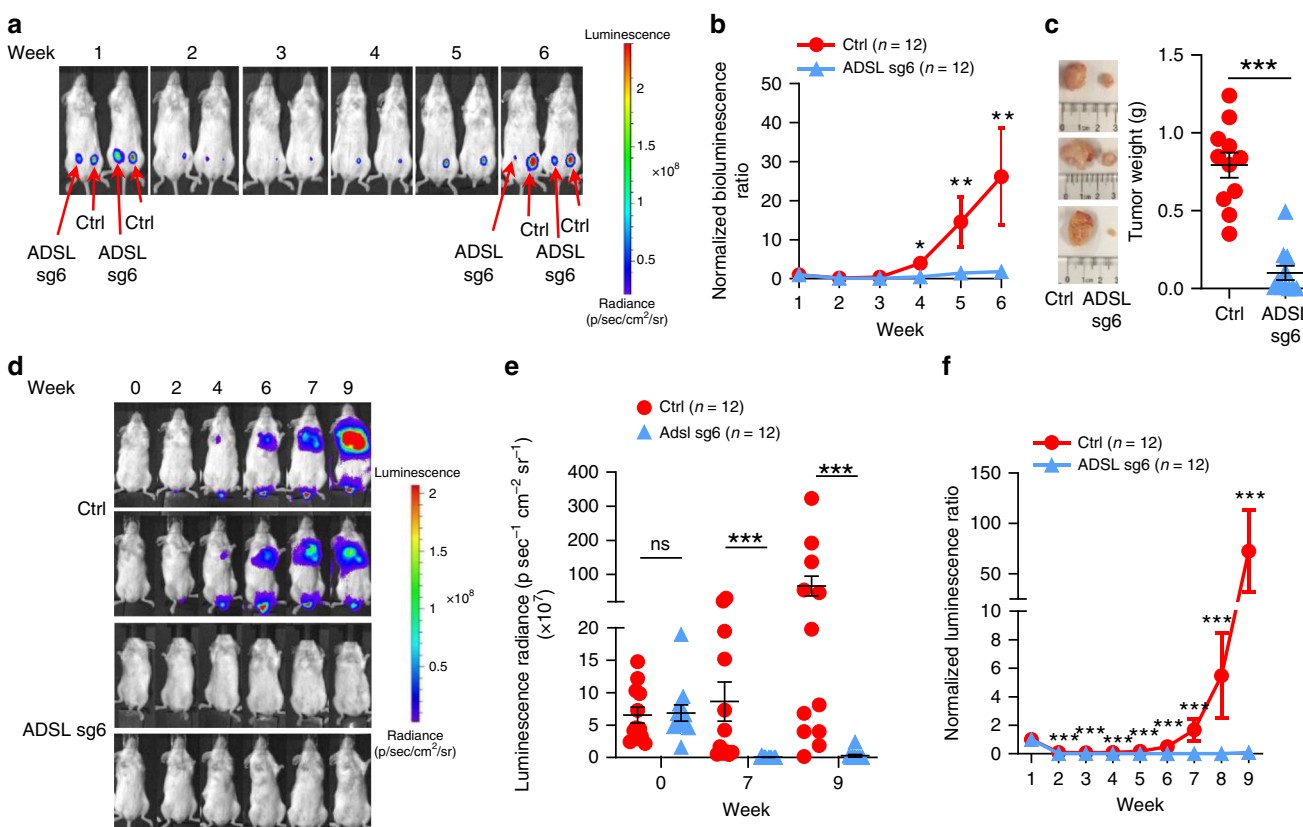

**Fig. 3** ADSL is required for TNBC tumorigenesis and lung colonization. **a** Representative bioluminescence images from the indicated weeks after the orthotopical injection of MDA-MB-231 luciferase-expressing cells infected with either ADSL sgRNA #6 (sg6) or control sgRNA (Ctrl) into the mammary fat pads of NOD SCID Gamma (NSG) mice. **b** Quantification of the bioluminescence imaging. **c** Representative images of tumors after dissection and quantification of tumor weight. **d** Representative bioluminescence images from the indicated weeks after the injection of MDA-MB-231 luciferase-expressing cells infected with either ADSL sgRNA #6 (sg6) or control sgRNA (Ctrl) into the tail vein of NSG mice. **e** Dot blot representation of the raw luminescence radiance in the indicated weeks after injection. **f** Quantification of the bioluminescence imaging. For (**b**), (**e**), and (**f**), the Mann-Whitney test was used to calculate the P values. For (**c**), two-tailed Student's t-test was used. Error bars represent SEM, *P < 0.05, **P < 0.01, ***P < 0.001. Source data are provided as a Source Data file

ADSL hydroxylation on proline 24 is important for maintaining ADSL enzymatic activity.

**ADSL regulates cMYC protein level through adenosine levels.** ADSL is an enzyme involved in purine metabolism. To examine the potential molecular mechanism by which ADSL contributes to TNBC tumorigenesis, we generated a profile of roughly 300 metabolites involved in several pathways, including nucleotide metabolism, oxidative phosphorylation, glycolysis, lipid metabolism, and the pentose-phosphate pathway. The levels of fumarate and AMP, products of ADSL enzymatic reactions, were significantly decreased upon ADSL depletion by two independent

sgRNAs (Supplementary Figs. 5a, b), validating the quality of our metabolomics mass spectrometry. We observed that the ADSL knockout effect was not limited to purine metabolism: other metabolic pathways were affected, including glutamate metabolism, the tricarboxylic acid cycle, and pyrimidine metabolism (Fig. 5a and Supplementary Fig. 5c). We also observed a drastic downregulation of both the mRNA and protein level of pyrimidine enzymes (Supplementary Fig. 5d). In order to understand the metabolic changes we observed, we performed ADSL gene expression profiling by RNA-seq followed by gene set enrichment analysis (GSEA). Interestingly, the oncogene cMYC signaling was among the pathways most affected by ADSL depletion in MDA-MB-231 cells (Fig. 5b, c), together with the E2F and TNFα

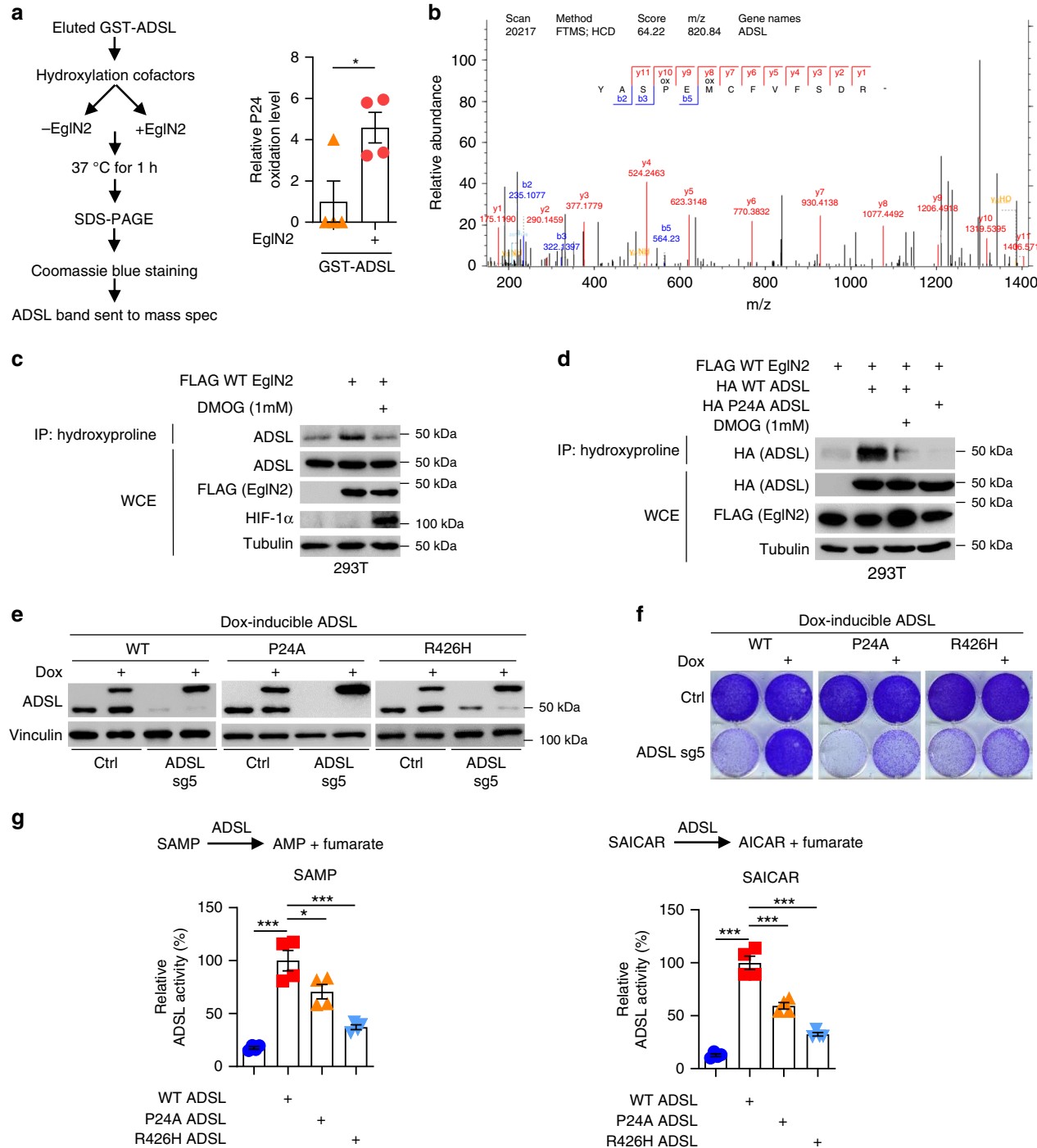

**Fig. 4** ADSL is hydroxylated by EglN2 on Proline 24. **a** In vitro hydroxylation of GST-ADSL in the presence or absence of recombinant EglN2. Bar graph represents the normalized ratio of the intensity of the oxidized P24-containing peptide to that of ADSL full protein. Error bars represent SEM, $n = 4$, *$P <$ 0.05 was calculated using two-tailed Student's $t$-test. **b** Fragmentation spectrum of proline hydroxylated peptide YASPEMCFVFSDR detected following experimental procedure described in (**a**). **c**, **d** IB of WCE and IP of lysates from 293T cells transfected with the indicated plasmids, and treated as indicated overnight. **e** IB of lysates from MDA-MB-231 cells overexpressing the indicated dox-inducible ADSL, transduced and treated as indicated. **f** Representative image of 2-D proliferation of MDA-MB-231 cells overexpressing the indicated dox-inducible ADSL, transduced and treated as indicated. **g** Enzymatic activity of WT, P24A or R426H ADSL in the presence of its substrate adenylosuccinate (SAMP) or succinylaminoimidazole carboxamide ribotide (SAICAR). Bar graphs represent the normalized percentage of ADSL activity compared with WT ADSL. Error bars represent SEM, $n = 4$, *$P <$ 0.05, **$P <$ 0.01, ***$P <$ 0.001 were calculated using one-way ANOVA followed by Tukey's multiple comparison test. Source data are provided as a Source Data file

signaling via NF-kB pathways (Supplementary Fig. 5e). We also checked the activation status of the mTORC pathway by analyzing mTORC substrate 4EBP1 phosphorylation: we found it greatly impaired (Supplementary Fig. 5f), in accordance with

previous reports linking purine metabolism with the activity of this important energy/nutrient/redox sensor[12,13].

Since cMYC has been reported to affect cell growth and proliferation by driving multiple metabolic pathways, including

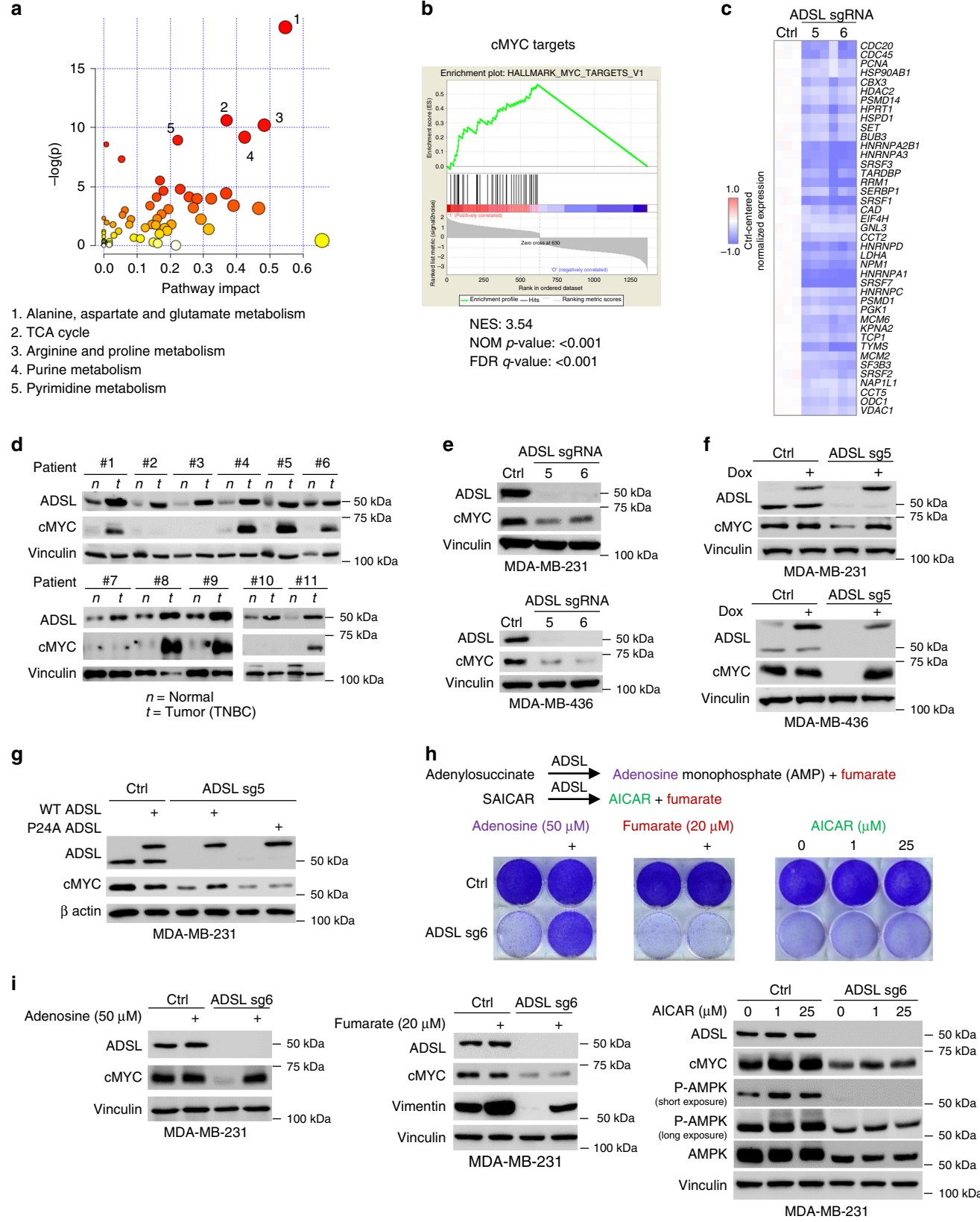

1. Alanine, aspartate and glutamate metabolism
2. TCA cycle
3. Arginine and proline metabolism
4. Purine metabolism
5. Pyrimidine metabolism

NES: 3.54
NOM *p*-value: <0.001
FDR *q*-value: <0.001

*n* = Normal
*t* = Tumor (TNBC)

the tricarboxylic acid cycle, glutamate and nucleotide metabolism[14], we speculated that ADSL might control cMYC expression in TNBC cells, which would explain, at least partially, the metabolic changes observed upon ADSL depletion. We first assayed the expression of cMYC in normal and TNBC tissues

from eleven patients, observing that in most cases, similarly to ADSL, cMYC was upregulated in TNBC (Fig. 5d). We also interrogated two datasets of TNBC patients, TCGA and METABRIC, about the correlation between *ADSL* and cMYC target gene expression. In accordance with our hypothesis, most

**Fig. 5** ADSL controls cMYC protein level by regulating adenosine levels. **a** Overview of the metabolic pathways affected by ADSL depletion. **b** Gene Set Enrichment Analysis (GSEA) of cMYC target genes between control and ADSL knockout MDA-MB-231 cells. **c** Heatmap of cMYC target gene expression upon ADSL depletion. **d** IB of lysates from paired TNBC patient-derived non-tumor (n) and tumor (t) breast tissues. **e** IB of lysates from MDA-MB-231 and MDA-MB-436 cells, transduced with indicated lentivirus. **f** IB of lysates from MDA-MB-231 and MDA-MB-436 cells overexpressing doxycycline (dox)-inducible ADSL, transduced with indicated lentivirus and treated as indicated. **g** IB of lysates from MDA-MB-231 cells overexpressing dox-inducible WT or P24A ADSL, transduced with indicated lentivirus. **h** Representative images of 2-D proliferation of MDA-MB-231 cells, transduced and treated as indicated. **i** IB of lysates from MDA-MB-231 cells, transduced with indicated lentivirus and treated as indicated. Source data are provided as a Source Data file

cMYC target genes were positively correlated with ADSL expression (Supplementary Table 3). We then depleted MDA-MB-231 and -436 TNBC cell lines with two independent ADSL sgRNAs and found a concomitant downregulation of cMYC upon ADSL depletion (Fig. 5e). This regulation was posttranscriptional, since no difference was detected on cMYC mRNA level (Supplementary Fig. 5g). Importantly, the regulation of ADSL sgRNA on cMYC was on-target since the CRISPR-Cas9 resistant version of WT ADSL could efficiently rescue the cMYC downregulation (Fig. 5f). To examine the effect of P24A mutation on cMYC regulation, we restored P24A ADSL in the ADSL knockout cell line. We did not observe the rescue of cMYC expression (Fig. 5g), suggesting that ADSL proline hydroxylation is critical for ADSL regulation of cMYC and breast cancer cell proliferation.

We then further investigated how the enzymatic activity of ADSL is responsible for the ADSL phenotype. We supplemented ADSL-depleted cells with the products of ADSL-catalyzed reactions, i.e., adenosine (50 μM), fumarate (20 μM), or AICAR (1 and 25 μM). Only adenosine was able to rescue TNBC cell proliferation (Fig. 5h). Fumarate was previously reported to promote the expression of vimentin, an epithelial-to-mesenchymal transition marker, in cancer cells[6]. Consistently, fumarate rescued the vimentin downregulation (Fig. 5i), but failed to rescue the growth defect (Fig. 5h) caused by ADSL depletion in MDA-MB-231. The phenotype was also not rescued by AICAR (Fig. 5h). We excluded the possibility that AICAR did not function by detecting the AICAR-induced phosphorylation of AMPK, as reported previously[15] (Fig. 5i). To verify that the rescue of ADSL depletion phenotype was specific to adenosine, we supplemented ADSL knockout cells with the other nucleosides. In fact, ADSL-depleted cells also showed decreased dCTP, dTTP and dGTP (Supplementary Fig. 5h). Thymidine (T), uridine (U), cytidine (C), guanosine (G), inosine (I), and the purine derivative hypoxanthine (H), all failed to restore cell growth (Supplementary Fig. 5i). Instead, the nucleobase adenine (Ade), similarly to adenosine (Ado), was able to rescue the ADSL depletion phenotype (Supplementary Fig. 5i). Consistently, we found that adenosine and adenine, but not fumarate and AICAR, rescued cMYC downregulation upon ADSL knockout in MDA-MB-231 cells (Fig. 5i and Supplementary Fig. 5j). Of note, adenosine could also rescue the expression of pyrimidine enzymes in a dose-dependent fashion (Supplementary Fig. 5k). Importantly, it seems that adenine and, by extension, adenosine do not need to be metabolized into AMP in order to rescue ADSL depletion phenotype: the downregulation of adenine phosphoribosyltransferase (APRT) (Supplementary Fig. 5l), in fact, was not able to prevent it (Supplementary Fig. 5m). Our results suggest that ADSL proline 24 hydroxylation is important for regulating ADSL activity, which controls adenosine (and adenine) levels that regulate cMYC protein level, and leads to downstream metabolic changes and breast cancer cell proliferation.

**ADSL inhibits cMYC negative regulator *MIR22HG* expression.** The fact that cMYC mRNA levels were unaffected by ADSL depletion suggested that ADSL somehow affected cMYC

translation or protein stability. The treatment of ADSL-depleted cells with the proteasome inhibitor MG132 induced an increase of cMYC protein level in all the samples, including the control (Supplementary Fig. 6a). This suggested that this proteasome-mediated mechanism is not specific to ADSL. Since a growing body of evidence points toward non-coding RNAs as strong posttranscriptional gene regulators[16–18], we searched for them in the RNA-seq-derived list of genes most significantly affected by ADSL depletion. Curiously, at the very top of the list was the long non-coding RNA *MIR22HG*, upregulated upon ADSL knockout (Fig. 6a, Supplementary Tables 4 and 5). We confirmed by qRT-PCR that ADSL negatively regulates *MIR22HG* in MDA-MB-231 (Fig. 6b) and MDA-MB-436 (Fig. 6c). Interrogating two different datasets of TNBC patients, TCGA and METABRIC, a negative correlation between *ADSL* and *MIR22HG* expression was found (Fig. 6d and Supplementary Fig. 6b). To investigate whether *MIR22HG* affects cMYC protein level, we overexpressed *MIR22HG* in MDA-MB-231 cells (Fig. 6e), and observed a decrease in cMYC protein level (Fig. 6f). Conversely, the downregulation of *MIR22HG* via three independent siRNAs (Fig. 6g) led to increased cMYC levels (Fig. 6h). In line with these data, we found a negative correlation between the mRNA expression of *MIR22HG* and that of several cMYC target genes in TNBC patients across two different datasets (Supplementary Table 6). We investigated whether *MIR22HG* is a key element for ADSL depletion phenotype, by creating a MDA-MB-231 cell line overexpressing *MIR22HG* in the presence of doxycycline. Similar to ADSL knockout, *MIR22HG* overexpression (Fig. 6i) impaired MDA-MB-231 cell 2-D (Fig. 6j) and 3-D colony formation (Fig. 6k), suggesting that *MIR22HG* is responsible, at least partially, for the cell growth defect observed in the absence of ADSL. Interrogating the link between ADSL and *MIR22HG* expression, we found that when re-introduced in ADSL-depleted cells, WT ADSL could restore *MIR22HG* expression (Fig. 6l). On the contrary, P24A mutant could not (Fig. 6m). Moreover, EglN2 depletion via two independent sgRNAs (Supplementary Fig. 6c) induced a significant increase of *MIR22HG* expression (Supplementary Fig. 6d) and, consistently, a decrease of cMYC protein level (Supplementary Fig. 6c). Collectively, these data indicate that ADSL-mediated regulation of *MIR22HG* expression requires ADSL hydroxylation by EglN2. We then wondered whether *MIR22HG* regulation, similar to cMYC, was also dependent on adenosine and adenine levels. Indeed, *MIR22HG* levels were restored to levels comparable to control in ADSL-depleted cells upon adenosine (Fig. 6n) or adenine (Supplementary Fig. 6e) treatment. Of note, neither adenosine nor adenine was able to alter *MIR22HG* levels in control cells, raising the hypothesis of a threshold, rather than a dose-dependent, mechanism for adenosine (and adenine) in regulating this long non-coding RNA. Our findings suggest that EglN2-mediated hydroxylation on ADSL proline 24 is important for modulating ADSL activity, which controls adenosine (and adenine) levels. Adenosine and adenine repress the long non-coding RNA *MIR22HG* expression, therefore upregulating cMYC protein level and inducing downstream metabolic changes and breast cancer cell growth (Fig. 6o).

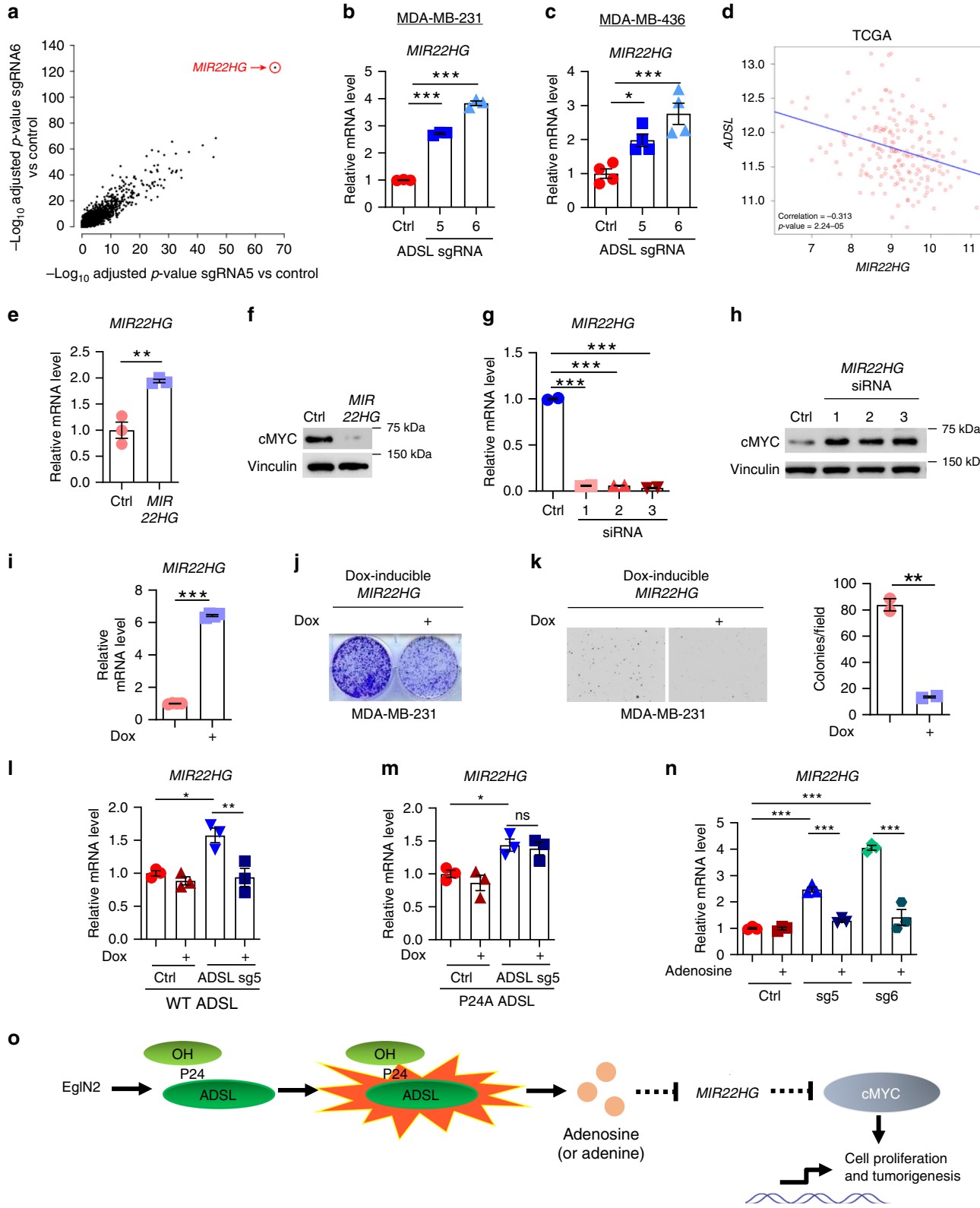

## Discussion

By developing an optimized enzyme-substrate-trapping strategy coupled with TAP-TAG purification and mass spectrometry, here we identify adenylosuccinate lyase (ADSL) as a substrate of the hydroxylase EglN2 in triple negative breast cancer (TNBC). Similar to EglN2 depletion, ADSL knockout impairs TNBC cell growth and invasion in vitro and in vivo. ADSL depletion, by affecting adenosine and adenine levels, leads to an increased expression of the long non-coding RNA *MIR22HG*. In turn, *MIR22HG* negatively regulates the oncogene cMYC protein level. When re-introduced in ADSL knockout cells, wild-type ADSL, but not the mutant missing the hydroxylation site (P24A), can rescue *MIR22HG* level, thus promoting oncogene cMYC expression and breast cancer cell growth.

**Fig. 6** ADSL controls cMYC negative regulator *MIR22HG* expression. **a** Scatterplot of differential-expression results: −log10 FDR-adjusted *P*-values from comparisons of each gRNA versus control using DESeq2 are plotted for all genes regardless of significance. **b, c** *MIR22HG* mRNA expression upon ADSL depletion in (**b**) MDA-MB-231 and (**c**) MDA-MB-436 cells. Graphs represent the mean ± SEM, *n* = 3 (**b**) and *n* = 4 (**c**). **d** Pearson correlation between the expression of *ADSL* and *MIR22HG* in TNBC patients from TCGA dataset. **e** *MIR22HG* mRNA expression and **f** cMYC protein level after transfecting MDA-MB-231 cells with either *MIR22HG* expressing plasmid or empty vector. **g** *MIR22HG* mRNA expression and **h** cMYC protein level after transfecting MDA-MB-231 cells with either three independent siRNAs targeting *MIR22HG* or siRNA control. **i** *MIR22HG* mRNA expression, **j** 2-D, and **k** 3-D colony formation in the presence or absence of doxycycline (dox) in MDA-MB-231 cells transduced with dox-inducible *MIR22HG* expressing lentivirus. Graphs represent the mean ± SEM from four independent sets of samples (**i**), and from two independent experiments, each performed in duplicate (**k**). **l, m** *MIR22HG* mRNA expression in ADSL control or knockout MDA-MB-231 cells with or without the expression of (**l**) WT and (**m**) P24A ADSL. **n** *MIR22HG* mRNA expression in ADSL control or knockout MDA-MB-231 cells treated as indicated (adenosine concentration was 50 μM). Graphs in (**l**), (**m**), and (**n**) represent the mean ± SEM, *n* = 3. **o** Schematic of the proposed mechanism by which EglN2-hydroxylated ADSL controls cMYC and cMYC target gene expression. *$P <$ 0.05, **$P <$ 0.01, ***$P <$ 0.001 were calculated using one-way ANOVA followed by Dunnett's multiple comparison test in (**b**), (**c**), and (**g**). In (**e**), (**i**), and (**k**), *P*-values were calculated using two-tailed Student's *t*-test. In (**l**), (**m**), and (**n**), *P*-values were calculated using one-way ANOVA followed by Tukey's multiple comparison test. Source data are provided as a Source Data file

The discovery of the mechanism of oxygen sensing has shed light on the importance of a previously understudied post-translational modification: protein hydroxylation[19,20]. Since then, great efforts have been made to investigate whether hydroxylases have substrates beyond HIF-α contributing to the physiological response to hypoxia. The main technical challenge is the transient nature of the hydroxylation reaction. Strategies capable of preventing the dissociation of the hydroxylase-substrate complex[4,21] have allowed the discovery of several substrates, including the kinase Akt[22], the phosphatase PP2A subunit B55α[23], the transcription factor FOXO3a[24], the kinases DYRK1A/B[25], the tumor suppressor p53[26], the erythropoietin receptor EPOR[27], and the kinase MAPK6[28]. Hydroxylation has been shown to have additional consequences than affecting protein stability, such as altering protein-protein interaction[22] and protein activity[22,28], and its dysregulation contributes to pathological outcomes[22,25]. Therefore, it is becoming clear that hydroxylation regulates cellular signaling to a much wider extent than initially thought, and uncovering its action may help understand pathological features and identify new potential therapeutic targets. Here we provide a strategy to identify hydroxylase substrates: the optimized substrate-trapping strategy, combining DMOG with hypoxia in HIF-1α-depleted cells, is followed by a TAP-TAG purification or, in parallel, a GST pull-down of the hydroxylase (here EglN2), and mass spectrometry. We chose to perform TAP-TAG purification to reduce the false positives: after two sequential affinity purifications, the chance of retrieving contaminants dramatically decreases. The proteins retrieved are further tested as potential substrates by comparing their interaction with the WT and catalytically dead (here EglN2-H358A) hydroxylase; if true substrates, they should display a stronger binding to the WT hydroxylase. This strategy has enabled us to identify ADSL as an EglN2 substrate.

ADSL has been studied in the context of ADSL deficiency, disorder characterized by a purine metabolism defect leading to several symptoms, including psychomotor retardation, microcephaly, and autistic features[29]. Even though ADSL activity has been previously reported to be dysregulated in breast cancer tissues[30], tubular and tubulo-villous adenoma[31], and gliomas[32], the mechanism underlying the potential role of ADSL in these malignancies has not been elucidated. Only recently a study provided a potential molecular mechanism for ADSL in endometrial cancer oncogenesis, by increasing killer cell lectin-like receptor C3 expression through fumarate production[33]. However, ADSL role in breast cancer, particularly TNBC, was completely unknown. Here we show that ADSL expression is significantly upregulated in TNBC patients compared with the other breast cancer subtypes and normal breast tissues. ADSL is essential for TNBC growth and invasiveness, as its depletion greatly impairs

both. Importantly, we show that the ADSL depletion phenotype is much less dramatic in normal breast epithelial cell lines, suggesting that a therapeutic window for a treatment targeting ADSL may be available. In a recently published study on "Project Drive"[34], a genome-wide shRNA screening identified ADSL as essential for cancer cell proliferation in 40 different cell lines, of which almost 25% were breast cancer cell lines. In comparison, the well-known oncogene PIK3CA was found critical in 29 cell lines, further strengthening the importance of studying ADSL in cancer.

As already observed for other hydroxylase substrates[22,25,28], ADSL hydroxylation does not affect protein stability. Instead, ADSL hydroxylation by EglN2 modulates ADSL activity. However, it remains unclear how ADSL proline 24 hydroxylation affects ADSL activity, which awaits further investigation. The hydroxylation may modify ADSL folding, enhancing its activity. Alternatively, it may alter the occurrence of other posttranslational modifications ultimately affecting ADSL activity, similar to EglN1-hydroxylated Akt[22] and FIH-hydroxylated RIPK4[28]. However, to the best of our knowledge, no posttranslational modifications of ADSL, apart from its hydroxylation shown here, are known so far. Another possibility is that hydroxylation may influence interactions between ADSL and other proteins affecting its activity. To address this question, in our future research we will examine the ADSL interactome in the presence or absence of prolyl hydroxylation.

Previous studies have highlighted the positive regulation of ADSL expression by cMYC[35,36]. However, the possibility of a feedback regulation has never been explored until now. We provide evidence that ADSL regulates cMYC expression. ADSL depletion leads to decreased cMYC protein levels and several cMYC target genes, which may explain, at least partially, the profound metabolic changes and overall phenotype observed upon ADSL knockout. Consistently, we show a positive correlation between *ADSL* and cMYC target genes expression in TNBC patients across two different datasets. A recent study showed that the cMYC mRNA 3′UTR is stabilized by adenosine, but could not provide any evidence for microRNA involvement in cMYC translation[37]. Here we find that the long non-coding RNA *MIR22HG* is overexpressed upon ADSL knockout, and its overexpression recapitulates, at least partially, ADSL depletion phenotype, in accordance with other reports that identified it as a tumor suppressor[38–40]. We demonstrate that *MIR22HG* expression is regulated by ADSL through adenosine and adenine, without the need to be metabolized into AMP. The identification of the molecular mechanism linking adenosine and adenine to the modulation of *MIR22HG* expression will be the object of our future research. We show that *MIR22HG* negatively regulates cMYC protein level and provide evidence of a negative

correlation between *MIR22HG* and cMYC target gene expression in TNBC patients in two different datasets. Consistently, a previous study showed that in breast cancer cell lines cMYC-binding protein MYCBP, a positive regulator of cMYC, is directly targeted by miR-22, leading to the downregulation of several E-box-containing cMYC target genes[41]. However, the fact that no change in cMYC protein level was observed[41] suggests that there may be an additional regulatory mechanism, likely directly ascribed to *MIR22HG*. It is worth noting that *MIR22HG* regulation of cMYC activity appears to be tissue-specific. In fact, in several lung cancer cell lines, *MIR22HG* downregulation causes a decrease in cMYC protein level[40]. Moreover, the regulation of MYCBP expression by the mature sequence miR-22-3p was not observed in hepatocellular carcinoma[39]. However, the exact mechanism by which *MIR22HG* modulates cMYC expression in TNBC has not been investigated here, and remains to be determined.

In conclusion, this study identifies the EglN2-hydroxylated ADSL as an important driver of TNBC proliferation and invasion both in vitro and in vivo. ADSL negatively controls the long non-coding RNA *MIR22HG* through a mechanism involving adenosine and adenine. *MIR22HG*, in turn, negatively regulates the oncogene cMYC expression. Therefore, the EglN2-ADSL-cMYC axis may be a potential therapeutic target in TNBC.

## Methods

**Cell culture and reagents**. MDA-MB-231, MDA-MB-436, MCF7, and 293T cells were cultured in DMEM supplemented with 10% fetal bovine serum (FBS) and 1% penicillin streptomycin (Pen Strep), T47D and MDA-MB-468 cells in 10% FBS, 1% Pen Strep RPMI, HMLE and MCF10A in MEGM (Lonza). All cell lines except 293T were obtained from ATCC. 293T were obtained from UNC Tissue Culture Facility, and authenticated via short tandem repeat testing. All cells were tested for mycoplasma contamination. Cells were maintained at 37 °C in a 5% $CO_2$ incubator. For hypoxia treatment, cells were maintained overnight in a 1% $O_2$ chamber (Coy Laboratory Products). Following lentivirus infection, cells were cultured in the presence of hygromycin (200 µg/ml), G418 (800 µg/ml), blasticidin (10 µg/ml), or puromycin (2 µg/ml) depending on the vector. Doxycycline (D9891), diethyl fumarate (D95654), adenosine (A4036), thymidine (T1895), cytidine (C4654), uridine (U3750), inosine (I4125), hypoxanthine (H9377), and guanosine (G6264) were from Sigma-Aldrich, DMOG (D1070-1g) from Frontier Scientific, MG132 (IZL-3175-v) from Peptide International, AICAR (sc-200659) and adenine (sc-291834) from Santa Cruz Biotechnology.

**Plasmids**. pcDNA-3.1-FLAG-EglN2-WT and -H358A were described previously[24]. pcDNA-3.1-HA-WT-ADSL, pTRIPZ-sg5 PAM mutated ADSL (WT, P24A, R426H), pGEX4T1-ADSL (WT, P24A), pGEX6P1-EglN1 and -EglN2, pGEX4T1-EglN3, pLKO APRT shRNA, pINDUCER20 *MIR22HG* were constructed by standard molecular biology techniques. The lentiviral vector containing full-length human *MIR22HG* gene was kindly provided by Dong-Yan Zhang and De-Hua Wu. Quick Change XL Site-Directed Mutagenesis Kit (200516, Agilent Technologies) was used to obtain ADSL mutants (P24A, R426H, silent mutation of sg5-PAM).

**Lentiviral sgRNA vectors, sgRNAs, shRNAs, and siRNAs**. CRISPR control and ADSL knockout were obtained by infecting cells with viruses produced using LentiCRISPR V2 plasmids. Ctrl and ADSL shRNA were obtained using pLKO plasmids. *MIR22HG*-targeting siRNAs were obtained from Dharmacon (RU-015037-00-0002 2 nmol). Target sequences are listed as follows:

Ctrl sgRNA: GCGAGGTATTCGGCGTCCGCG
ADSL sgRNA #5: TGTGCTTCGTGTTTAGCGAC
ADSL sgRNA #6: ACAGGTATAAATTCCGGACA
Ctrl shRNA: AACAGTCGCGTTTGCGACTGG
ADSL shRNA #2: CTCACTGCCACAGAGTATAAT
ADSL shRNA #4: GAAGGTGAAAGCAGAATTATG
HIF-1α shRNA: CCAGTTATGATTGTGAAGTTA
APRT shRNA: CCACTCTGTGGGCCTCCTATT
EglN2 sgRNA #13: AGAGGTGGCTGTGGCTCTGG
EglN2 sgRNA #14: GCAGCGCCTTCACTCTGCAG

**Virus production and infection**. 293T packaging cell line was used for lentivirus production. Following transfection of the packaging plasmids and the lentivirus-based plasmid with lipofectamine 3000 (Invitrogen) according to the manufacturer instructions, viruses were collected after 48 h. Once filtered through a 0.45 µM, an appropriate volume of viruses was used to infect target cells in the presence of 8 µg/ml polybrene. Subsequently, target cell lines underwent the antibiotic selection dictated by the plasmid.

**Western blot analysis and antibodies**. Cells were lysed using EBC buffer (50 mM Tris pH 8.0, 120 mM NaCl, 0.5% NP-40, 0.1 mM EDTA and 10% Glycerol) supplemented with complete protease and phosphatase inhibitors (Roche Applied Biosciences). Fresh-frozen samples of TNBC and their adjacent normal tissues were obtained from tissue procurement facility at UNC. All human tumor sample-related studies have been approved by the University of North Carolina at Chapel Hill Institutional Review Board (IRB). Proteins from human tissues were extracted using RIPA buffer (Sigma-Aldrich). Lysate concentrations were measured by Bradford assay. Equal amounts of cell lysates were resolved by SDS-PAGE. Rabbit antibody against ADSL (HPA000525, dilution 1:1000) and mouse antibodies against vinculin (V9131, dilution 1:20000) and α-tubulin (T9026, dilution 1:5000) were from Sigma-Aldrich. Rabbit anti-hydroxyproline (ab37067) and APRT (ab91428, dilution 1:250) antibodies were from Abcam. Rabbit antibodies against HIF-1α (14179), cMYC (5605), RRM1 (8637), 4EBP1 (9644), AMPK (5832), P-AMPK (2535), P-4EBP1 (2855), FLAG (14793), HA (3724), and GST (2625) tags were from Cell Signaling Technology, and were all diluted 1:1000. Mouse antibody against HA tag (901501, dilution 1:1000) was from BioLegend. Mouse antibody against vimentin (550513, dilution 1:1000) was from BD Biosciences. Rabbit antibody against EglN2 (NB100-310, dilution 1:2000) was from Novus Biologicals. Mouse antibodies against RRM2 (sc-376973, dilution 1:500), CAD (sc-376072, dilution 1:1000), TYMS (sc-390945 dilution 1:1000), DHODH (sc-166348, dilution 1:1000), GMPS (sc-376163, dilution 1:1000), and β actin (sc-47778, dilution 1:1000) were from Santa Cruz Biotechnology. Rabbit antibody against CTPS1 (A304-543A, dilution 1:1000), peroxidase conjugated goat anti-mouse (31430, dilution 1:5000) and anti-rabbit (31460, dilution 1:5000) secondary antibodies were purchased from Thermo Fisher Scientific. Uncropped scans of the most important blots have been provided in the Source Data file.

**Immunoprecipitation**. The cell lysates were clarified by centrifugation and then mixed with anti-hydroxyproline, anti-EglN2 (Abcam, ab113077) or anti-ADSL (Abcam, ab151958) antibodies, anti-FLAG M2 affinity gel (Sigma-Aldrich) or 3F10 HA conjugated beads (Roche Applied Bioscience) overnight. For the antibodies against hydroxyproline, EglN2 and ADSL, cell lysates were further incubated with protein G sepharose beads (Roche Applied Bioscience) for 3 h. Bound complexes were washed with NETN buffer eight times and were eluted by boiling in SDS loading buffer. Bound proteins were resolved in SDS-PAGE followed by western blot analysis.

**TAP-TAG purification**. Cells were washed once with ice-cold PBS, then scraped from the p150 plate in 5 ml of ice-cold PBS per plate. Cells were collected and washed for a second time with ice-cold PBS. An appropriate volume of lysis buffer (50 mM Tris-HCl pH 7.5, 150 mM NaCl, 0.5% NP-40, 10% glycerol) was added to obtain a final protein concentration of about 10 µg/µl. The lysates were clarified by centrifugation, mixed with anti-FLAG M2 affinity gel (Sigma-Aldrich) and then incubated under rotation for 4 h at 4 °C. Beads were washed for five times with lysis buffer, then FLAG-tagged proteins were eluted by incubating the beads with 0.5 mg/ml FLAG peptide (Sigma-Aldrich, F3290) in Pierce centrifuge columns (Thermo Fisher Scientific, 89898) for 30 min at 4 °C under gentle rotation. FLAG-tagged protein elution was repeated for a total of three times. The elution was then mixed with 3F10 HA conjugated beads (Roche Applied Bioscience) and incubated under rotation overnight at 4 °C. HA beads were washed for five times with lysis buffer, then HA-tagged proteins were eluted by incubating HA beads with 8 M urea buffer (200 mM Tris-HCl pH 7.5, 100 mM NaCl) for 30 min at room temperature under constant rotation. HA-tagged protein elution was repeated for a total of three times. The final elution was filtered, then ice-cold 100% trichloroacetic acid (TCA) was added to the elution to obtain a final concentration of 20% TCA. After centrifugation at 4 °C at 20,000 × g, the pellet was washed with ice-cold 10% TCA once, then three times with ice-cold acetone. Every wash was followed by a 30-min centrifugation at 4 °C at 20,000 × g. Finally, the pellet was let air-dry at room temperature, and sent to mass spectrometry analysis.

**In vitro hydroxylation**. Comparable amounts of eluted GST-ADSL (WT and P24A) proteins were supplemented with 50 mM HEPES (pH 7.4), 1500 U/mL catalase, 100 µM $FeSO_4$, 5 mM ascorbic acid, 1 mM α-KG, and 2 µg of purified recombinant EglN2 (or the same volume of its buffer, as negative control) in a 100-µl reaction volume. Following a 1-h incubation at 37 °C, the beads were washed, digested with trypsin as described previously[42] and analyzed by LC-MS/MS. Alternatively, 20 µL of each reaction underwent the IP with hydroxyproline antibody.

**GST protein purification and GST pull-down**. BL21 competent cells were transformed with GST plasmids. Single colonies were cultured in 50 ml LB medium containing ampicillin. After overnight culture, 15 ml were diluted in 200 ml LB medium for shaking at 37 °C for 2-3 h until OD600 was about 0.8–1.0. IPTG (0.2 mM) was added to induce GST protein production for 4 h (or overnight at

25 °C for GST-ADSL) before harvesting pellets. Bacteria lysates were disrupted by sonication. Cleared bacteria lysates were purified by using Glutathione-sepharose 4B beads. Twenty microliters of GST suspension proteins were incubated with either in vitro translated protein in 500 µl NETN buffer or cell lysates. After overnight incubation, bound complexes were washed with NETN buffer eight times followed by boiling in SDS loading buffer and SDS-PAGE.

**RNA-Seq analysis**. Total RNA from triplicates was extracted by using RNeasy kit with on column DNase digestion (Qiagen). The RNA-Seq library was prepared using TruSeq RNA Library Prep Kit v2 (Illumina) according to the manufacturer's instructions. Sequencing was performed at the High Throughput Genomic Sequencing facility of UNC. Samples were sequenced on an Illumina HiSeq2500 with paired-end 50 bp reads. Reads were then trimmed and filtered of adapter sequencing using cutadapt, and required to have at least 90% of bases with quality scores exceeding 20. Reads were then aligned to the reference human genome (hg19) using STAR[43], and transcript abundance was estimated using salmon[44]. Differential expression between each gRNA and control was analyzed using DESeq2[45]. GSEA was then run using variance-stabilizing-transformed expression values.

**Real-time RT-PCR**. Total RNA was isolated with RNeasy mini kit (Qiagen). First strand cDNA was generated with the iScript cDNA synthesis kit (Biorad). Real-time PCR was performed using iTaq universal SYBR Green Supermix (BioRad) following the manufacturer's instruction and was conducted in triplicate. Real-time RT-PCR primers used were as follows:

*MIR22HG* F = CGAACAGCAGGGTGGATGAT
*MIR22HG* R = CGCACTATGGTGCCACATCT
Tubulin F = GCTGTGGAAAACCCAAGAAGC
Tubulin R = AAGTTCGCACTGGCACCTAC
*RPL32* F = GCTGCTGATGTGCAACAAA
*RPL32* R = GGGATTGGTGACTCTGATGG
*CMYC* F = TGAGGAGACACCGCCCAC
*CMYC* R = CAACATCGATTTCTTCCTCATCTTC

**Cell proliferation assay**. Cells were seeded, in duplicate, in 6-well plates ($10^5$ cells/well) in appropriate growth medium. Medium was changed every 2 days. For the rescue experiments with nucleotides, AICAR and fumarate, treatment was performed just after seeding the cells, and was renewed every day. After 7 days, cells were fixed with 4% formaldehyde for 10 min at room temperature, stained for 10 min with 0.5% crystal violet and then washed several times with distilled water. Once dried, the plates were scanned.

**Soft-agar colony formation assay**. Cells were plated at a density of 5000 cells per ml in complete medium with 0.4% agarose, onto bottom layers composed of medium with 1% agarose followed by incubation at 4 °C for 10 min. Afterward, cells were moved to 37 °C incubator. Every 4 days, 200 µl of complete media were added onto the plate. After 4 weeks, the extra liquid on the plate was withdrawn and 1 ml medium supplemented with 100 µg/ml iodonitrotetrazoliuim chloride solution was added onto each well. After incubating overnight at 37 °C, the colonies were counted under the microscope.

**Cell invasion assay**. MDA-MB-231 and MDA-MB-468 cell invasion assay was performed using BD BioCoat Matrigel Invasion Chamber (354480) according to the manufacturer's instructions. In total, $3 \times 10^4$ (for MDA-MB-231) and $3 \times 10^5$ (for MDA-MB-468) cells were inoculated into each chamber in triplicate and incubated for 18 h at 37 °C, 5% $CO_2$ incubator. The cells on the lower surface of the membrane were stained using Diff-Quick stain kit (B4132-1A) from SIEMENS, and then counted under EVOS XL Core Microscope (Cat# AMEX1000, Thermo Fisher Scientific).

**Orthotopic tumor growth**. Six-week old female NOD SCID Gamma mice (NSG, Jackson lab) were used for xenograft studies. Approximately $6 \times 10^5$ viable MDA-MB-231 cancer cells expressing luciferase or $5 \times 10^5$ viable MDA-MB-468 were resuspended in 40 µl matrigel (Corning, 354234) and injected orthotopically and bilaterally into the mammary fat pad of each mouse. For bioluminescent detection and quantification of cancer cells, mice were given a single intraperitoneal injection of luciferin (150 mg/kg) in sterile Dulbecco's phosphate-buffered saline. Fifteen minutes later, mice were placed in a light-tight chamber equipped with a charge-coupled device IVIS imaging camera (Xenogen, Alameda, CA). Photons were collected for a period of 1–60 s, and images were obtained and quantified using LIVING IMAGE 2.60.1 software (Xenogen). The total photons from the ADSL sgRNA tumor region of interest (ROI) were normalized to the photons of the first imaging for each mouse. Results were presented as mean ± standard error of the mean (SEM). Mice were sacrificed 6 weeks (for MDA-MB-231) or 11 weeks (for MDA-MB-468) after the first imaging. The rough mass of tumors was presented as mean ± SEM. All animal experiments were in compliance with National Institutes of Health guidelines and received ethical approval by the University of North Carolina at Chapel Hill Animal Care and Use Committee.

**Breast cancer lung colonization**. Six-week-old female NOD SCID Gamma mice (NSG, Jackson lab) were used for xenograft studies. Approximately $3 \times 10^5$ viable luciferase-labeled MDA-MB-231 cancer cells were resuspended in 50 µl PBS and injected into the tail vain. Bioluminescence imaging was performed as described in the section above. Mice were sacrificed 8 weeks (for ADSL sg5) or 9 weeks (ADSL sg6) after the first imaging. All animal experiments were in compliance with National Institutes of Health guidelines and received ethical approval by the University of North Carolina at Chapel Hill Animal Care and Use Committee.

**Hydroxylation site identification by mass spectrometry**. 293T cells were transfected with either HA-tagged ADSL or empty vector, then treated with either 1 mM DMOG or control overnight. HA affinity gel was used to pull-down HA-tagged ADSL. The samples were digested with trypsin and processed as previously described[42]. Desalted peptides were analyzed on a Q-Exactive mass spectrometer (Thermo, Germany).

**ADSL activity assay**. ADSL activity was measured as described previously[46]. Briefly, HPLC analysis of AMP and AICAR formed from both ADSL substrates, SAMP (Sigma-Aldrich) and SAICAR (synthesized as described in ref. [47]), was performed. Reactions were run for 20 min at 37 °C in 65 µl ADSL buffer and 20 ng protein. The substrate concentrations were 0.25 and 0.15 mM for SAMP and SAICAR, respectively.

**dNTP extraction from cells and measurement**. dNTPs were extracted from cells as follows: $2 \times 10^6$ cells were washed twice with DPBS, and scraped in 400 µl of ice-cold 60% methanol. Samples were vortexed vigorously to lyse the cells and then heated at 95 °C for 3 min, prior to centrifugation at $12,000 \times g$ for 3 min. The supernatants were collected and completely vacuum-dried. The dried pellets were subsequently resuspended in dNTP buffer (50 mM Tris-HCl, pH 8.0 and 10 mM $MgCl_2$). 1–2 µl of the extracted dNTP samples were used for each 20 µl single nucleotide incorporation assay, which was performed as described previously[48].

**Metabolite extraction from cells and metabolomics analysis**. In total, $5 \times 10^5$ cells were seeded onto 6-well plates in quadruplicate. Next day, the culture medium was completely removed, cells were immediately placed on dry ice, followed by the addition of 1 ml 80% methanol/water (pre-cooled in −80 °C freezer) to each well. After incubation in −80 °C freezer for 15 min, cells were scraped into 80% methanol on dry ice, transferred to Eppendorf tubes, and centrifuged at $20,000 \times g$ for 10 min at 4 °C. The supernatant was split into two Eppendorf tubes before speed-vacuum drying. The samples were then sent to Locasale group for meta-bolomics analysis. The dry pellets were reconstituted into 30 µL sample solvent (water:methanol:acetonitrile, 2:1:1, v/v) and 3 µl were further analyzed by liquid chromatography-mass spectrometry (LC-MS). Ultimate 3000 UHPLC (Dionex) was coupled to Q Exactive Plus-Mass spectrometer (QE-MS, Thermo Scientific) for metabolite profiling. A hydrophilic interaction chromatography method (HILIC) employing an Xbridge amide column (100 × 2.1 mm i.d., 3.5 µm; Waters) was used for polar metabolite separation. Detailed LC method was described previously[49] except that mobile phase A was replaced with water containing 5 mM ammonium acetate (pH 6.8). The QE-MS was equipped with a HESI probe with related parameters set as below: heater temperature, 120 °C; sheath gas, 30; auxiliary gas, 10; sweep gas, 3; spray voltage, 3.0 kV for the positive mode and 2.5 kV for the negative mode; capillary temperature, 320 °C; S-lens, 55; A scan range (m/z) of 70–900 was used in positive mode from 1.31 to 12.5 min. For negative mode, a scan range of 70–900 was used from 1.31 to 6.6 min and then 100–1000 from 6.61 to 12.5 min; resolution: 70,000; automated gain control (AGC), $3 \times 10^6$ ions. Customized mass calibration was performed before data acquisition. LC-MS peak extraction and integration were performed using commercial available software Sieve 2.2 (Thermo Scientific). The integrated peak area was used to represent the relative abundance of each metabolite in different samples. The missing values were handled as described in a previous study[49].

**U-$^{13}C_6$ glucose isotopomer analysis**. In total, $5 \times 10^5$ cells were seeded onto 6-well plates in quadruplicate in regular high glucose (4.5 g/L) DMEM, supplemented with 10% FBS and 1% pen/strep. Next day, the cells were washed twice with PBS, and the medium was replaced with glucose-free DMEM containing 10% FBS, 1% pen/strep, and 4.5 g/L [U-$^{13}C_6$] glucose. After 24 h, the metabolites were extracted and the metabolomics analysis performed as described in the previous paragraph.

**Study approval**. All animal studies were approved by the UNC Institutional Animal Care and Use Committee.

**Statistics**. The unpaired two-tailed Student's *t*-test was used for experiments comparing two sets of data. For experiments comparing more than two conditions, differences were tested by a one-way ANOVA followed by Dunnett's or Tukey's multiple comparison tests. Mann-Whitney test was used for analyzing the animal experiments. Data represent mean ± SEM from at least two independent replicates. *, ** and *** denote *P* value of < 0.05, 0.01, and 0.005, respectively. NS denotes not significant. All values were calculated with Prism 5 (GraphPad).

**Reporting summary**. Further information on research design is available in the Nature Research Reporting Summary linked to this article.

## Data availability

The RNA-seq dataset generated for this study has been deposited to the GEO with Study Accession number GSE136414. The mass spectrometry proteomics data for substrate-trapping interaction analysis have been deposited to the ProteomeXchange Consortium via the PRIDE[50] partner repository with the dataset identifiers PXD015787 and PXD015790. The mass spectrometry proteomics data for ADSL hydroxyproline analysis have been deposited to the ProteomeXchange Consortium via the PRIDE[50] partner repository with the dataset identifier PXD015773. All other relevant data supporting the main findings of the present study are available throughout the article, its Supplementary Information files and the source data file or from the corresponding author upon reasonable request. The source data underlying Fig. 1d–i, 2b–h, 3b, c, e, f, 4a, c–e, g, 5d–g, i, 6b, c, e–i, k–n, Supplementary Figures 1d, e, f, 2b–j, 3 b–d, 4 a–e, 5 a, b, d, f–h, j–l and 6a, c–e are provided as a Source Data File.

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

## Acknowledgements

We thank all members of the Zhang and Perou laboratories for helpful discussions and suggestions; Dong-Yan Zhang and De-Hua Wu for providing *MIR22HG* overexpression plasmid; UNC Proteomics Center (particularly Laura Herring), UNC Tissue Procurement Facility, and UNC Animal Studies Core. Q.Z. was supported by The V Foundation Scholar Award, University Cancer Research Fund, Mary Kay Foundation, National Cancer Institute (Q.Z., R01CA211732, R21CA223675), Kimmel Scholar Award, and Susan G. Komen Career Catalyst Award. A.v.K. was supported by Wellcome Trust (Multiuser Equipment Grant, 208402/Z/17/Z). J.M.S. and T.S.P. were supported by The Eunice Kennedy Shriver National Institute of Child Health and Human Development (U54HD079124) and NINDS (P30NS045892). J.M.H. and B.K. were supported by NIH (R01AI136581). M.Z. was supported by Charles University [programmes PRIMUS/17/MED/6 and PROGRES Q26/LF1] and by the Ministry of Education, Youth and Sports of CR [LQ1604 National Sustainability Programme II]. This work was also supported in part by Cancer Prevention and Research Institute of Texas (CPRIT, RP190058 to Q.Z.).

## Author contributions

Q.Z. conceived the study. G.Z. and Q.Z. planned the experiments and analyzed the data. G.Z., X.L., M.T., A.R. and J.Z. performed experiments. C.F. performed the patient data analysis. J.M.S. and T.S.P. provided the RNA-seq bioinformatics analysis. J.R. performed the mass spectrometry analysis of the hydroxylation sites under the supervision of A.v.K. J.L. performed the metabolomics and isotopomer analysis under the supervision of J.W.L. J.M.H. and B.K. measured cell dNTPs. M.Z. and B.J. independently measured ADSL activity. L.X. and X.C. provided help with mass spectrometry analysis. M.L. provided help on obtaining TNBC patient samples. C.M.P. helped to supervise the study and provided critical advice. The paper was written by G.Z. and Q.Z. with input and editing from all authors.

## Competing interests

The authors declare no competing interests.
