## [Peer Review File · Nature Communications]

Reviewers' comments:

Reviewer #1 (Remarks to the Author):

The manuscript from Zurlo and colleagues reports that the prolyl hydroxylase EglN2 posttranslationally modified adenylosuccinate lyase (ADSL) in Triple Negative Breast Cancer (TNBC). Using a substrate trapping strategy based on the combination of hypoxia stress and PHD inhibition (DMOG), the authors report that ADSL is hydroxylated by EglN2 on proline 24 and this modification is required for breast cancer cell proliferation. Not surprisingly, the authors demonstrate that loss or knockdown of ADSL using shRNA or CRISPR Cas 9 led to decrease in cancer cell proliferation and tumor growth. The authors showed that the effects of ADSL loss on cell proliferation can be salvaged by providing exogenous adenosine in the media. The authors demonstrate that ADSL loss leads to an increase in MIR22HG, a microRNA that seems to target c-MYC expression. Additionally, MIR22HG levels can be rescued by re-expressing wild-type ADSL but not by expressing a non hydroxylatable ADSL.

This study is very interesting and technically well performed. However, to acquire a better understanding of the mechanistic effects of ADSL regulation by EglN2 in breast cancer, some comments should be addressed:

Major comments:

- 1) According to the authors, the effects of ADSL loss can be rescued by providing exogenous adenosine, but not by addition of fumarate, AICAR, thymidine, uridine, cytidine and guanosine. The authors claim that the antiproliferative effects induced by loss of ADSL are mediated by a decrease in cellular adenosine levels, but this interpretation appears incorrect. Certainly, the authors showed that inhibition of cell growth upon ADSL loss is adenosine dependent, however, adenosine can be converted into nucleotides through the purine salvage pathway sustaining nucleotide synthesis required to make RNA, DNA and maintaining cellular homeostasis. If the authors use inosine instead of adenosine, I predict that the outcome will be the same, inosine (or hypoxanthine, adenine), like adenosine, should also be able to rescue ADSL effects. The effects described by the authors as specifically adenosine dependent should be confirmed, the authors should assess inosine (or/and hypoxanthine, adenine) effects. Moreover, the authors should knock-down or knockout the purine salvage enzymes (APRT and HPRT) to assess whether the nucleosides (adenosine, inosine etc.) can still rescue the antiproliferative effects of ADSL loss and whether adenosine (or inosine) need to be metabolized to mediate the rescue?
- 2) While the authors demonstrated alteration in ADSL activity in vitro between WT and mutant ADSL, the authors should perform an isotopic tracing experiment using ¹³C-glycine or ¹⁵N-glutamine to measure the activity of this pathway in cells expressing ADSL WT or ADSL P24A.
- 3) The rescue of MYC protein levels by adenosine in ADSL knockout system is interesting, but it is unlikely that this effect is restricted to MYC or that MYC downregulation can fully explain the decrease in cancer cell growth. Purine depletion induced by ADSL loss changes global translation or transcription. The authors should look at other key transcription factors or signaling pathways, for example mTORC1 signaling...
- 4) ADSL is a bona fide MYC target. The authors suggest that ADSL, through the regulation of MIR22HG expression by adenosine levels, can reciprocally control MYC expression. This proposed model is interesting but quite puzzling. The regulation of MYC by ADSL raises the question on whether MIR22HG alone can explain the regulation of MYC protein levels downstream of ADSL. The authors should address the question on whether inosine, adenine or hypoxanthine can control MIR22HG and MYC protein levels?
- 5) One key question that the authors should address is whether MIR22HG surexpression can recapitulate the effects of ADSL loss on breast cancer cell growth?

Minor comments:

The abstract should mention the residue hydroxylated (P24) by EGLN2 and the biological effects when this proline is mutated to alanine.

Panel 5I: Similarly to MYC, protein levels of AMPK seem to decrease in response to ADSL loss. The authors should explain this. Can this be rescued by exogenous nucleosides?

Reviewer #2 (Remarks to the Author):

The study by Qing Zhang and the colleagues reported the discovery and functional study of a novel proline hydroxylation target ADSL. They developed a more efficient trapping-based enrichment workflow using both hypoxia and DMOG treatment to stabilize transient interaction between EGLN2 and its targets. Using both in vivo and in vitro tagging system, they performed co-immunoprecipitation and mass spectrometry analysis. ADSL was identified as the only common EGLN2 targets from the two workflows in the triple negative breast cancer cells. Functional study with xenograft model showed that ADSL expression was up-regulated in breast cancer and its activity was required for breast cancer cell proliferation. Through in vitro and in vivo assays, the authors confirmed that ADSL was a proline hydroxylation substrate with a major site on Pro24. Hydroxylation at proline 24 promoted ADSL activity in nucleotide metabolism. Very interestingly, through RNA-seq analysis, the authors found that inhibition of ADSL activity decreased c-Myc expression in breast cancer cells and induced the expression of a long non-coding RNA MIR22HG. Authors further demonstrated that MIR22HG was a negative regulator of c-Myc. The overall study is exciting and the data are quite significant. The new trapping strategy for prolyl hydroxylase substrate analysis will be very useful to identify new proline hydroxylation-dependent pathways. Characterizations of ADSL and MIR22HG as new regulator of c-Myc in triple negative breast cancer cells are important findings. However, some technical and conceptual discussions still need further clarification. The manuscript may be suitable for publication after addressing the following concerns.

Major concerns:

1. It is quite surprising that ADSL was the only target identified by TAP-TAG and GST IP studies. Are there other known EGLN2 target identified? Maybe the criterion to select a target was too stringent or the efficiency of the workflow needs further improvements.
2. If ADSL was likely the major target of EGLN2 in the triple negative breast cancer cells, could the authors check if EGLN2 regulates TNBC or c-Myc through ADSL activity or its downstream MIR22HG?
3. One mass spectrum was provided for the identification of P24 hydroxylation on ADSL (Fig. 4B). However, the quality of the data was poor and key fragment ions were ambiguously assigned, making it difficult to distinguish a possible Met double oxidation and proline hydroxylation. It is highly recommended that the data should be re-analyzed with confident assignment.
4. As evidence to demonstrate that ADSL proline 24 is the major target of EGLN2, the authors have performed in vitro hydroxylation assay and quantification. To further demonstrate that EGLN2 is the major enzyme that regulates ADSL proline hydroxylation, it will be very helpful to perform in vivo overexpression and knockdown of EGLN2 and confirm its role in regulating endogenous ADSL modification.

Minor concerns:

1. Fig 2A showed that ADSL expression levels were elevated in TNBC, but in the protein level blots in Fig 2B, it seems that protein levels of ADSL were significantly different between normal and tumor tissues. Is this because the ADSL protein is not stable and is regulated at the posttranslational level? In addition, are the data in Fig 2A statistically significant? If so, they should be labeled.
2. Fig S4A showed that DMOG treatment decreased the P24 proline hydroxylation level on ADSL in TNBC, but it lacks an error bar, so it is hard to evaluate its significance. It also seems that DMOG was not efficient in reducing the hydroxylation level of ADSL.
3. The quality of Fig 2A and Fig. 4B needs improvement. The fonts could not be recognized.

Reviewer #3 (Remarks to the Author):

Zurlo et al. utilize a novel mechanism to enhance, or trap, the hydroxylase (Egln2) with its substrates in order to detect binding proteins using mass spectrometry. They identified ADSL as a novel egln2 substrate in overexpression or in vitro experiments in which HIF-1 is depleted. The group further shows in multiple models that ADSL knockout impairs 2D and 3D cell proliferation and xenograft tumor growth. They show that hydroxylated-ADSL induces cMYC by regulating adenosine levels in a mechanism that requires miR22HG expression. The experiments were well thought out and conducted in multiple models including overexpression, depleted, and re-expression experiments.

I have three main concerns for consideration:

All of the Egln2-ADSL interaction experiments are conducted with either purified GST-Egln2 protein or cells overexpressing a FLAG-tag version of Egln2. Have the authors tried to conduct pull down experiments with endogenous Egln2? If they cannot detect endogenous binding this should at least be mentioned in the text. It does raise concerns that the binding is an artifact of overexpression. Likewise all hydroxylation experiments were conducted in the context of overexpressed ADSL or using in vitro translated protein.

The authors should show that Egln2 decreases cMyc protein levels in accordance with their overall scheme in Figure 6L.

Can the authors comment on why the addition of adenosine alone does not alter miR22HG levels (Figure 6K). Why is this specific to only rescuing ADSL deficient adenosine production?

Reviewer #4 (Remarks to the Author):

In this manuscript, Zurlo and colleagues provide very interesting data showing that ADSL (adenylosuccinate lyase) is a bona fide Egln2 prolyl hydroxylase substrate and it promotes aggressive properties of TNBC via activating cMYC signaling. The proposed model is conceptually novel and most (though not all) data is solid and supports the conclusions made by the authors.

Major points:

1. Figure 1E: In contrast to what's stated in the Text (page 6, bottom), IP with mutant FLAG-Egln2-H358A clearly pulled down ADSL in MDA-MB-231 cells (lane 3).
2. Figure 2: Data in this figure has many issues.
 - In Fig. 2A about ADSL mRNA levels, the Y-axis needs to be more clearly labeled and the N for each subtype of breast cancer and P values in comparison with "Normal" indicated in the dot plots. Importantly, in the UNC337 and METABRIC datasets, the ADSL mRNA levels in "LumA and/or LumB" seem to be lower than in "Normal", which should be explained.
 - Do increased ADSL mRNA levels in TNBC correlate with worse patient survival?
 - Fig. 2C: sgRNA-6 in MCF-10A and HMLE cells clearly reduced cell numbers, in contrast to authors' statement "..... did not find overt growth defect..... (page 8, line 5)".
 - Fig. 2D and Fig. 2G: the colony formation images are not that good.
3. Figure 3 (ADSL is required for TNBC tumorigenesis and metastasis): These are important in vivo assays that drive a point essential to the entire project; thus, similar experiments in another TNBC (preferably a PDX) model should be carried out. Critically, experiments in Fig. 3D-F did not measure spontaneous metastasis but rather measure the TNBC cell trapping/survival/proliferation in the lung. How come the images shown in Fig. 3D did not reveal any luc. signal but the dot plots

in Fig. 3E showed significant readings in some animals?

4. Figure 4: The mass-spec histogram in Fig. 4A needs to be better presented.

5. Figure 5/Fig. S5 and related: Fig. 5A needs to be more clearly labeled and Fig. 5C needs to be re-scaled so that the Ctrl lanes can show differences in gene expression. Importantly, data in these figures clearly suggests rather 'pleiotropic' effects of ADSL depletion on TNBC cell metabolism (purine/pyrimidine/glutamate metabolism and TCA cycle), raising the critical question of how authors can really separate such general effects vs. the effect on cMYC. Data shown in Table S1 is problematic: Most 'Correlation values' are pretty low (in correlation studies the Correlation Coefficients are more important than p-values). What type of correlation study was performed. Presentations of data in Fig. 5H-I in the Text were mixed up.

6. Figure 6/Fig. S6 and related (ADSL-MIR22HG-MYC): There exists reasonably strong evidence in the literature about MIR22HG functioning as a tumor suppressor. Data shown in Fig. 6D/Fig. S6B/Table S2 is problematic: Most 'Correlation values' are pretty low and it's not clear which type of correlation study was performed. Critically, it is unclear how ADSL, not a transcription factor, negatively regulates MIR22HG in TNBC cells, NOR is it clear how MIR22HG negatively regulates cMYC.

Point to Point Response

Reviewer #1

The manuscript from Zurlo and colleagues reports that the prolyl hydroxylase EgIN2 posttranslationally modified adenylosuccinate lyase (ADSL) in Triple Negative Breast Cancer (TNBC). Using a substrate trapping strategy based on the combination of hypoxia stress and PHD inhibition (DMOG), the authors report that ADSL is hydroxylated by EgIN2 on proline 24 and this modification is required for breast cancer cell proliferation. Not surprisingly, the authors demonstrate that loss or knockdown of ADSL using shRNA or CRISPR Cas 9 led to decrease in cancer cell proliferation and tumor growth. The authors showed that the effects of ADSL loss on cell proliferation can be salvaged by providing exogenous adenosine in the media. The authors demonstrate that ADSL loss leads to an increase in MIR22HG, a microRNA that seems to target c-MYC expression. Additionally, MIR22HG levels can be rescued by re-expressing wild-type ADSL but not by expressing a non hydroxylatable ADSL.

This study is very interesting and technically well performed. However, to acquire a better understanding of the mechanistic effects of ADSL regulation by EgIN2 in breast cancer, some comments should be addressed:

Major comments:

1) According to the authors, the effects of ADSL loss can be rescued by providing exogenous adenosine, but not by addition of fumarate, AICAR, thymidine, uridine, cytidine and guanosine. The authors claim that the antiproliferative effects induced by loss of ADSL are mediated by a decrease in cellular adenosine levels, but this interpretation appears incorrect. Certainly, the authors showed that inhibition of cell growth upon ADSL loss is adenosine dependent, however, adenosine can be converted into nucleotides through the purine salvage pathway sustaining nucleotide synthesis required to make RNA, DNA and maintaining cellular homeostasis. If the authors use inosine instead of adenosine, I predict that the outcome will be the same, inosine (or hypoxanthine, adenine), like adenosine, should also be able to rescue ADSL effects. The effects described by the authors as specifically adenosine dependent should be confirmed, the authors should assess inosine (or/and hypoxanthine, adenine) effects. Moreover, the authors should knock-down or knockout the purine salvage enzymes (APRT and HPRT) to assess whether the nucleosides (adenosine, inosine etc.) can still rescue the antiproliferative effects of ADSL loss and whether adenosine (or inosine) need to be metabolized to mediate the rescue?

Response: We thank the reviewer for raising this important point. We assessed inosine, adenine and hypoxanthine potential ability to rescue ADSL-depleted cells: only adenine was able to rescue the phenotype (**Supplementary fig. S5i**).

In addition, as suggested, we determined the effect of APRT downregulation in ADSL-depleted cells treated with exogenous adenosine (and adenine). We did not perform the HPRT downregulation since neither inosine nor hypoxanthine was able to rescue the phenotype. We show that adenosine and adenine do not need to be metabolized in order to rescue the ADSL-depletion phenotype (**Supplementary figs. S5l and S5m**).

2) While the authors demonstrated alteration in ADSL activity in vitro between WT and mutant ADSL, the authors should perform an isotopic tracing experiment using ^{13}C -glycine or ^{15}N -glutamine to measure the activity of this pathway in cells expressing ADSL WT or ADSL P24A.

Response: We performed an isotopic experiment using $[\text{U-}^{13}\text{C}_6]$ glucose in control and ADSL-depleted cells expressing either ADSL WT or P24A with the help from our collaborator Dr. Jason Locasale. AMP is the direct product of ADSL-catalyzed reaction. Only ADSL-depleted cells expressing WT ADSL displayed an isotope incorporation in AMP comparable to control cells. On the contrary, AMP isotope incorporation in cells expressing P24A ADSL was similar to ADSL knockout cells (**Supplementary Fig. S4e**).

3) The rescue of MYC protein levels by adenosine in ADSL knockout system is interesting, but it is unlikely that this effect is restricted to MYC or that MYC downregulation can fully explain the decrease in cancer cell growth. Purine depletion induced by ADSL loss changes global translation or transcription. The authors should look at other key transcription factors or signaling pathways, for example mTORC1 signaling.

Response: We agree with the reviewer that the severe phenotype upon ADSL depletion is very unlikely determined by MYC downregulation alone. As suggested, we checked a broader spectrum of potentially affected transcription factors and signaling pathways. Specifically, we show from our transcriptomics analysis that E2F1 downstream signaling, similarly to MYC, is negatively affected upon ADSL depletion (**Supplementary Fig. S5e**). On the contrary, TNF α signaling via NF- κ B is upregulated in ADSL-depleted cells (**Supplementary Fig. S5e**).

Due to the space limitation, we could not include the respective heat maps, which appear as follows:

In addition, we measured E2F1 protein level, which appears reduced upon ADSL depletion (result not shown in the revised paper), and examined mTORC1 signaling by assessing the phosphorylation of mTORC1 downstream effector 4EBP1, which is also negatively affected by ADSL knockout (**Supplementary Fig. S5f**).

4) ADSL is a bona fide MYC target. The authors suggest that ADSL, through the regulation of MIR22HG expression by adenosine levels, can reciprocally control MYC expression. This proposed model is interesting but quite puzzling. The regulation of MYC by ADSL raises the question on whether MIR22HG alone can explain the regulation of MYC protein levels downstream of ADSL. The authors should address the question on whether inosine, adenine or hypoxanthine can control MIR22HG and MYC protein levels?

Response: MIR22HG alone seems to be able to regulate MYC protein levels downstream of ADSL, since its overexpression leads to a decrease in MYC protein amount (**Fig. 6f**), whereas its downregulation causes an increase in MYC levels (**Fig. 6h**).

F

H

From **figs 5i and 6n**, it appears that adenosine can control MYC and MIR22HG expression, respectively. This regulation can be seen only in ADSL-depleted cells, possibly because in tumor cells with control sgRNA, the nucleoside levels are already high enough.

We added this possible explanation in the revised main text (**Page 16**): “Of note, neither adenosine nor adenine was able to alter MIR22HG levels in control cells, raising the hypothesis of a threshold, rather than a dose-dependent, mechanism for adenosine (and adenine) in regulating this long non-coding RNA.”).

As suggested, we assessed whether inosine, adenine or hypoxanthine can control MIR22HG and MYC levels. As expected from the phenotype results (inability to rescue ADSL depletion growth defect), inosine and hypoxanthine did not control MIR22HG and MYC levels (**results not shown in the paper due to the space limitation**):

On the contrary, adenine could restore both MIR22HG (**Supplementary Fig. S6g**) and MYC (**Supplementary Fig. S5j**) levels.

5) One key question that the authors should address is whether MIR22HG surexpression can recapitulate the effects of ADSL loss on breast cancer cell growth?

Response: We thank the reviewer for this very important question. We created a dox-inducible *MIR22HG* overexpressing TNBC cells (MDA-MB-231) and analyzed the phenotype. Upon *MIR22HG* overexpression, the 2-D and 3-D colony formation was greatly impaired (**Figs. 6i, 6j and 6k**), suggesting that the effects of ADSL depletion are at least partially mediated by *MIR22HG* overexpression.

Minor comments:

The abstract should mention the residue hydroxylated (P24) by EgIN2 and the biological effects when this proline is mutated to alanine.

Response: Thanks for the suggestion. We modified the abstract accordingly. “Mechanistically, an integrated transcriptomics and metabolomics analysis revealed that ADSL promotes the activation of the oncogenic cMYC pathway by regulating cMYC protein level via a mechanism requiring ADSL hydroxylation **on Proline 24 (Page 2)**. Specifically, **hydroxylation-proficient ADSL (Page 2)**, by affecting adenosine levels, represses the expression of the long non-coding RNA MIR22HG, thus upregulating the oncogene cMYC protein level and its target gene expression. Our findings identify the critical role of ADSL hydroxylation in controlling cMYC and TNBC tumorigenesis.”

Panel 5I: Similarly to MYC, protein levels of AMPK seem to decrease in response to ADSL loss. The authors should explain this. Can this be rescued by exogenous nucleosides?

Response: Our RNA-seq data show that PRKAA1 and PRKAG2 genes, coding for a catalytic and regulatory subunit of AMPK, respectively, are significantly downregulated upon ADSL depletion, which could explain AMPK decreased protein amount in ADSL-depleted cells (**results not shown due to the space limitation**):

We checked whether this can be rescued by exogenous nucleosides, and we did not observe a restoration of AMPK levels (**results not shown due to the space limitation**):

We acknowledged these observations in the main text, as follows (**Pages 13-14**):

“We ruled out the possibility that AICAR did not efficiently enter the cells by detecting the AICAR-induced phosphorylation of AMPK, as reported in a previous study (14) (Fig. 5I). Of note, AMPK level appeared greatly impaired upon ADSL knockout. This downregulation, which could not be rescued by exogenous adenosine, was likely due to the significant decrease of the expression of the genes encoding the AMPK catalytic subunit alpha 1 (PRKAA1) and non-catalytic regulatory subunit gamma 2 (PRKAG2), as revealed by the RNA-seq analysis (data not shown).”

Reviewer #2

The study by Qing Zhang and the colleagues reported the discovery and functional study of a novel proline hydroxylation target ADSL. They developed a more efficient trapping-based enrichment workflow using both hypoxia and DMOG treatment to stabilize transient interaction between EGLN2 and its targets. Using both in vivo and in vitro tagging system, they performed co-immunoprecipitation and mass spectrometry analysis. ADSL was identified as the only common EGLN2 targets from the two workflows in the triple negative breast cancer cells. Functional study with xenograft model showed that ADSL expression was up-regulated in breast cancer and its activity was required for breast cancer

cell proliferation. Through in vitro and in vivo assays, the authors confirmed that ADSL was a proline hydroxylation substrate with a major site on Pro24. Hydroxylation at proline 24 promoted ADSL activity in nucleotide metabolism. Very interestingly, through RNA-seq analysis, the authors found that inhibition of ADSL activity decreased c-Myc expression in breast cancer cells and induced the expression of a long non-coding RNA MIR22HG. Authors further demonstrated that MIR22HG was a negative regulator of c-Myc.

The overall study is exciting and the data are quite significant. The new trapping strategy for prolyl hydroxylase substrate analysis will be very useful to identify new proline hydroxylation-dependent pathways. Characterizations of ADSL and MIR22HG as new regulator of c-Myc in triple negative breast cancer cells are important findings. However, some technical and conceptual discussions still need further clarification. The manuscript may be suitable for publication after addressing the following concerns.

Major concerns:

1. It is quite surprising that ADSL was the only target identified by TAP-TAG and GST IP studies. Are there other known EGLN2 target identified? Maybe the criterion to select a target was too stringent or the efficiency of the workflow needs further improvements.

Response: This, to our knowledge, is the first attempt to identify EglN2 substrates in triple negative breast cancer. Therefore, unfortunately, we cannot compare the result of our screen with previously identified EGLN2 substrates in other setting. In addition, this is the first attempt, as we are aware of, to look for other EglN substrates in the absence of HIF. We would like to stress that we purposively make our approach very stringent in order to avoid the identification of false positive substrates. We underlined our choice of a stringent approach in the revised manuscript (**Page 18**):

“We chose to perform TAP-TAG purification to reduce the probability of false positives: in fact, after two sequential affinity purifications, the chance of retrieving contaminants dramatically decreases.”

2. If ADSL was likely the major target of EGLN2 in the triple negative breast cancer cells, could the authors check if EGLN2 regulates TNBC or c-Myc through ADSL activity or its downstream MIR22HG?

Response: We thank the reviewer for raising this important question. We checked whether EGLN2 regulates TNBC through ADSL. We overexpressed ADSL in control and EglN2-depleted MDA-MB-231 cells (**Supplementary Fig. S1g**). ADSL overexpression, even if

moderate, could partially restore MDA-MB-231 2-D colony formation (**Supplementary Fig. S1h**), partially due to the possibility that EglN2 may regulate other pathways (beside ADSL) that contribute to the phenotype in these cells.

3. One mass spectrum was provided for the identification of P24 hydroxylation on ADSL (Fig. 4B). However, the quality of the data was poor and key fragment ions were ambiguously assigned, making it difficult to distinguish a possible Met double oxidation and proline hydroxylation. It is highly recommended that the data should be re-analyzed with confident assignment.

Response: There are two y-ions that allow to distinguish the m(diox) from the m(ox)+p(hyd), y8 and y9. We see the ions for the m(ox)+p(hyd), but none for the proposed m(diox). Mass accuracy of the fragments is unequivocal.

b			y	y ²	b		y	y ²			
---	1	Y	13	---	---	1	Y	13			
235.1077	2	A	12	1477.6086	739.3080	235.1077	2	A	12	1477.6086	739.3080
322.1397	3	S	11	1406.5715	703.7894	322.1397	3	S	11	1406.5715	703.7894
419.1925	4	P	10	1319.5395	660.2734	435.1874	4	P(Hyd)	10	1319.5395	660.2734
548.2351	5	E	9	1222.4867	611.7470	564.2300	5	E	9	1206.4918	603.7495
711.2654	6	M(Diox)	8	1093.4441	547.2257	711.2654	6	M(Ox)	8	1077.4492	539.2282
871.2961	7	C(Carba)	7	930.4138	465.7105	871.2961	7	C(Carba)	7	930.4138	465.7105
1018.3645	8	F	6	770.3832	385.6952	1018.3645	8	F	6	770.3832	385.6952
1117.4329	9	V	5	623.3148	312.1610	1117.4329	9	V	5	623.3148	312.1610
1264.5013	10	F	4	524.2463	262.6268	1264.5013	10	F	4	524.2463	262.6268
1351.5333	11	S	3	377.1779	189.0926	1351.5333	11	S	3	377.1779	189.0926
1466.5603	12	D	2	290.1459	145.5766	1466.5603	12	D	2	290.1459	145.5766
---	13	R	1	175.1190	88.0631	---	13	R	1	175.1190	88.0631

In addition, we can show with these following narrower spectra that we have a match for the p(hyd)m(ox) but none for the m(diox).

Raw File giada1 Scan 20217 Method FTMS; HCD Score 64.22 m/z 820.84 Gene names ADSL

- Y A S P E O M C F V F F S D R -
 b b b

Raw File giada1 Scan 20217 Method FTMS; HCD Score 64.22 m/z 820.84 Gene names ADSL

- Y A S P E O M C F V F F S D R -
 b b b

We replaced Fig 4B with the following higher quality figure:

4. As evidence to demonstrate that ADSL proline 24 is the major target of EGLN2, the authors have performed in vitro hydroxylation assay and quantification. To further demonstrate that EGLN2 is the major enzyme that regulates ADSL proline hydroxylation, it will be very helpful to perform in vivo overexpression and knockdown of EGLN2 and confirm its role in regulating endogenous ADSL modification.

Response: We performed an in vivo overexpression of EglN2 and checked the hydroxyproline level of endogenous ADSL, which increases. Moreover, we show that the hydroxylase inhibitor DMOG prevents the increase of ADSL hydroxyproline level observed upon EglN2 overexpression (**Fig. 4c**).

Minor concerns:

1. Fig 2A showed that ADSL expression levels were elevated in TNBC, but in the protein level blots in Fig 2B, it seems that protein levels of ADSL were significantly different between normal and tumor tissues. Is this because the ADSL protein is not stable and is regulated at the posttranslational level? In addition, are the data in Fig 2A statistically significant? If so, they should be labeled.

Response: The variation between different normal and tumor samples may reflect tissue heterogeneity from different patients, which we acknowledge in the revised main text (**Page 8**) (“Despite the variation observed between different normal and tumor samples, which may reflect tissue heterogeneity from different patients, ADSL was always upregulated in tumors compared to their adjacent normal tissue (Fig. 2B)”). Our key point is to show that there is increased level of ADSL protein in TNBC tumor compared to adjacent normal.

The data in Fig 2A are statistically significant. We have generated the following new figure in which the statistical significance, together with the sample size, is labeled:

2. Fig S4A showed that DMOG treatment decreased the P24 proline hydroxylation level on ADSL in TNBC, but it lacks an error bar, so it is hard to evaluate its significance. It also seems that DMOG was not efficient in reducing the hydroxylation level of ADSL.

Response: We generated evidence that DMOG was efficient in reducing the hydroxylation level of ADSL P24 also in a parallel mass spec experiment, showed in the revised paper in Fig S4A. As the reviewer kindly pointed out, the absence of error bar in Fig S4A could have generated confusion, so we decided to not show it in the revised paper.

Importantly, we provide in the revised paper another piece of data supporting our mass spec results: DMOG treatment drastically reduces endogenous ADSL proline hydroxylation observed upon EglN2 overexpression (**Fig. 4c**).

3. The quality of Fig 2A and Fig. 4B needs improvement. The fonts could not be recognized.

Response: We improved the quality of Fig2A and 4B, which now appear as follows:

Reviewer #3:

Zurlo et al. utilize a novel mechanism to enhance, or trap, the hydroxylase (Egln2) with its substrates in order to detect binding proteins using mass spectrometry. They identified ADSL as a novel egln2 substrate in overexpression or in vitro experiments in which HIF-1 is depleted. The group further shows in multiple models that ADSL knockout impairs 2D and 3D cell proliferation and xenograft tumor growth. They show that hydroxylated-ADSL induces cMYC by regulating adenosine levels in a mechanism that requires miR22HG expression. The experiments were well thought out and conducted in multiple models including overexpression, depleted, and re-expression experiments.

I have three main concerns for consideration:

1) All of the Egln2-ADSL interaction experiments are conducted with either purified GST-Egln2 protein or cells overexpressing a FLAG-tag version of Egln2. Have the authors tried to conduct pull down experiments with endogenous Egln2? If they cannot detect endogenous binding this should at least be mentioned in the text. It does raise concerns that the binding is an artifact of overexpression. Likewise all hydroxylation experiments were conducted in the context of overexpressed ADSL or using in vitro translated protein.

Response: We thank the reviewer for raising this important point. We performed endogenous IP of ADSL and Egln2 in order to verify their interaction in a physiologically relevant setting. We show that endogenous ADSL and Egln2 IP can retrieve Egln2 and ADSL, respectively (**Fig. 1f**).

2) The authors should show that Egln2 decreases cMyc protein levels in accordance with their overall scheme in Figure 6L.

Response: We downregulated Egln2 by two independent sgRNAs and checked cMYC protein level, which indeed decreases (**Supplementary Fig. S6e**).

3) Can the authors comment on why the addition of adenosine alone does not alter miRH22HG levels (Figure 6K). Why is this specific to only rescuing ADSL deficient adenosine production?

Response: We think that the levels of adenosine in TNBC cells with control sgRNA are already high enough to modulate MIR22HG expression; hence, the addition of more adenosine does not produce visible alteration. We added this potential explanation in the text (**Page 16**):

“Of note, neither adenosine nor adenine was able to alter *MIR22HG* levels in control cells, raising the hypothesis of a threshold -rather than a dose-dependent- mechanism for adenosine (and adenine) in regulating this long non-coding RNA.”

Reviewer #4:

In this manuscript, Zurlo and colleagues provide very interesting data showing that ADSL (adenylosuccinate lyase) is a bona fide EgIN2 prolyl hydroxylase substrate and it promotes aggressive properties of TNBC via activating cMYC signaling. The proposed model is conceptually novel and most (though not all) data is solid and supports the conclusions made by the authors.

Major points:

1. Figure 1E: In contrast to what’s stated in the Text (page 6, bottom), IP with mutant FLAGEgIN2-H358A clearly pulled down ADSL in MDA-MB-231 cells (lane 3).

Response: Thanks for pointing out that sentence, which could have misled the readers. We modified the text as follows (**Page 7**):

“The binding of wild-type EgIN2 with ADSL was much stronger than that of the catalytically inactive H358A mutant, suggesting that not only is ADSL an interactor, but also a substrate, of EgIN2 (Fig. 1E)”.

2. Figure 2: Data in this figure has many issues.

- In Fig. 2A about ADSL mRNA levels, the Y-axis needs to be more clearly labeled and the N for each subtype of breast cancer and P values in comparison with “Normal” indicated in the dot plots.

Response: We thank the reviewer for the suggestion. We replaced the figure with the following which addresses the raised points:

We also present the comparison between TNBC and true normal tissues in TCGA dataset (UNC337 and METABRIC datasets do not contain information about true normal tissues) (Fig. 2b):

Importantly, in the UNC337 and METABRIC datasets, the ADSL mRNA levels in “LumA and/or LumB” seem to be lower than in “Normal”, which should be explained.

Response: We apologized with the reviewer for the unclear labeling. In the unrevised paper, fig. 2A displayed normal-like, not true normal tissue values. We have corrected this misleading labeling in the revised figure:

- Do increased ADSL mRNA levels in TNBC correlate with worse patient survival?

Response: We have performed several different survival analysis (including COX Proportional Hazards Model, Concordance Index and K-M analysis) and found that ADSL mRNA level does not correlate with patient survival in TNBC (**Supplementary Fig. S2a**). We discuss this in the revised paper (**Page 8**):

“However, by performing the survival analysis of TNBC patients in TCGA, UNC337 and METABRIC (including Kaplan-Meier curves, COX Proportional Hazard Model and Concordance Index), no correlation was found between ADSL mRNA level and patient survival (Supplementary Fig. S2A). In this regard, we would like to underline that it has been widely acknowledged that the prognosis data are neither necessary nor sufficient to predict whether the target gene is a good therapeutic target in cancer (Kaelin, Common pitfalls in preclinical cancer target validation, Nat Rev Cancer, 2017).

- Fig. 2C: sgRNA-6 in MCF-10A and HMLE cells clearly reduced cell numbers, in contrast to authors’ statement “..... did not find overt growth defect..... (page 8, line 5)”.

Response: Thanks for pointing out this inaccuracy. We modified the text as follows (**Page 8**):

“To test whether ADSL may be important for normal breast epithelial cell proliferation, we also depleted ADSL in MCF-10A and HMLE cells: although also these cell lines showed a growth defect, this was not as dramatic as that observed in TNBC cell lines”.

- Fig. 2D and Fig. 2G: the colony formation images are not that good.

Response: We provided better images in the revised figures 2D and 2G, which now appear as follows:

3. Figure 3 (ADSL is required for TNBC tumorigenesis and metastasis): These are important in vivo assays that drive a point essential to the entire project; thus, similar experiments in another TNBC (preferably a PDX) model should be carried out.

Response: We performed an additional in vivo experiment depleting ADSL with the two independent guides sg5 and sg6 in another TNBC cell line, MDA-MB-468. We show that ADSL depletion impairs tumor growth also in this setting (**Supplementary Figs. 3c and 3d**):

- Critically, experiments in Fig. 3D-F did not measure spontaneous metastasis but rather measure the TNBC cell trapping/survival/proliferation in the lung.

Response: We thank the reviewer for highlighting this mislabeling. We corrected the text accordingly.

- How come the images shown in Fig. 3D did not reveal any luc. signal but the dot plots in Fig. 3E showed significant readings in some animals?

Response: This would lead to a saturated image for the Ctrl mice in the later time points. We included an excel sheet with the raw data in the supplemental excel sheet.

4. Figure 4: The mass-spec histogram in Fig. 4A needs to be better presented.

Response: We improved it as follows:

5. Figure 5/ Fig. S5 and related: Fig. 5A needs to be more clearly labeled and Fig. 5C needs to be re-scaled so that the Ctrl lanes can show differences in gene expression.

Response: We improved these figures as suggested.

- Importantly, data in these figures clearly suggests rather 'pleiotropic' effects of ADSL depletion on TNBC cell metabolism (purine/pyrimidine/glutamate metabolism and TCA cycle), raising the critical question of how authors can really separate such general effects vs. the effect on cMYC.

Response: We think that these general effects cannot be separated from the effect on cMYC. On the contrary, since cMYC is known to affect plenty of metabolic pathways, one possible explanation would be that ADSL-depletion broad effects are due, at least in part, to ADSL regulation on cMYC.

- Data shown in Table S1 is problematic: Most ‘Correlation values’ are pretty low (in correlation studies the Correlation Coefficients are more important than p-values). What type of correlation study was performed.

Response: The aim of this table was to show the positive trend correlating ADSL expression with cMYC target expression. Even though singular genes may have low correlation values, almost all the values are positive, and for some genes the correlation is pretty strong (e.g. CDC20, CDC45, HPRT1, TYMS). Pearson correlation was performed. We included this information in the revised text (**Supplementary Information, Page 4**).

- Presentations of data in Fig. 5H-I in the Text were mixed up.

Response: We corrected this mistake.

6. Figure 6/ Fig. S6 and related (ADSL-MIR22HG-MYC): There exists reasonably strong evidence in the literature about MIR22HG functioning as a tumor suppressor. Data shown in Fig. 6D/ Fig. S6B/ Table S2 is problematic: Most ‘Correlation values’ are pretty low and it’s not clear which type of correlation study was performed.

Response: We provide evidence that *MIR22HG* may function as a tumor suppressor in TNBC, since its overexpression drastically decreases MDA-MB-231 cell 2D and 3D colony formation (**Figs. 6i, 6j and 6k**):

For Fig. 6D, Fig. S6B, and Table S2, Pearson correlation was performed. We included this information in the revised text (**Page 43, Supplementary Information, Pages 3-4**).

- Critically, it is unclear how ADSL, not a transcription factor, negatively regulates MIR22HG in TNBC cells, NOR is it clear how MIR22HG negatively regulates cMYC.

Response: We thank the reviewer for this insightful comments. We have provided experimental evidence that ADSL, through regulating adenosine levels, regulates *MIR22HG* and *MYC*. We also performed extensive gain-of-function as well as loss-of-function experiments to corroborate this finding. However, this specific comment may be

beyond the scope of the current paper since our study is aimed at identifying novel EglN2 substrates that may play an important role in breast cancer. We added some discussion on this part in the revised text, which will be the aim of our future research (**Page 20**):

“The identification of the molecular mechanism linking adenosine and adenine to the modulation of *MIR22HG* expression will be the object of our future research.”

“However, the exact mechanism by which *MIR22HG* modulates cMYC expression in TNBC has not been investigated here, and remains to be determined.”

REVIEWERS' COMMENTS:

Reviewer #1 (Remarks to the Author):

The authors have addressed most of my comments.

Reviewer #2 (Remarks to the Author):

The authors have done a great job and satisfactorily addressed the concerns raised by this reviewer. For data availability, raw MS data and database search results acquired for substrate-trapping interaction analysis and ADSL hydroxyproline identification should be deposited in a public database such as ProteomeXchange. The manuscript is recommended for publication.

Reviewer #3 (Remarks to the Author):

The authors have substantially improved the manuscript and have adequately addressed my concerns in the revised version.

Daniele Gilkes

Reviewer #4 (Remarks to the Author):

The authors have made conscientious efforts to address most of my comments.

POINT BY POINT RESPONSE TO REVIEWERS:

Reviewer #1 (Remarks to the Author):

The authors have addressed most of my comments.

Response: We thank the reviewer for the insightful comments.

Reviewer #2 (Remarks to the Author):

The authors have done a great job and satisfactorily addressed the concerns raised by this reviewer. For data availability, raw MS data and database search results acquired for substrate-trapping interaction analysis and ADSL hydroxyproline identification should be deposited in a public database such as ProteomeXchange. The manuscript is recommended for publication.

Response: We thank the reviewer for the great suggestions. The mass spectrometry proteomics data for substrate-trapping interaction analysis have been deposited to the ProteomeXchange Consortium via the PRIDE partner repository with the dataset identifiers PXD015787 and PXD015790. The mass spectrometry proteomics data for ADSL hydroxyproline analysis have been deposited to the ProteomeXchange Consortium via the PRIDE partner repository with the dataset identifier PXD015773. We have included these references into the Data Availability section in the revised manuscript.

Reviewer #3 (Remarks to the Author):

The authors have substantially improved the manuscript and have adequately addressed my concerns in the revised version.

Daniele Gilkes

Response: We thank the reviewer for her constructive comments.

Reviewer #4 (Remarks to the Author):

The authors have made conscientious efforts to address most of my comments.

Response: We thank the reviewer for the important questions raised.